# Mode Connectivity in Unlearning: A Loss Landscape Analysis of Machine Unlearning

## Abstract

Machine Unlearning aims to remove undesired information from trained models without requiring full retraining from scratch. Despite recent advancements, their underlying loss landscapes and optimization dynamics received less attention. In this paper, we investigate and analyze machine unlearning through the lens of mode connectivity–the phenomenon where independently trained models can be connected by smooth low-loss paths in the parameter space. We define and study mode connectivity in unlearning (MCU) across a range of overlooked conditions, including models trained curriculum learning, second-order optimization, and cross-method connectivity. Our findings show distinct patterns of loss landscapes across various datasets, training paradigms, and unlearning methods. With MCU, we analyze the mechanistic (dis)similarity between unlearning methods. We also demonstrate MCU can be used to improve generalization of unlearning and defending against relearning attacks. To the best of our knowledge, this is the first study of loss landscape analysis of machine unlearning with mode connectivity.

## 1 Introduction

The widespread deployment of machine learning models raises the need for *machine unlearning*–the process of removing specific knowledge from a trained model without affecting other knowledge (Bourtoule et al., 2021a; Liu et al., 2024b). This need is driven by both legal and ethical imperatives, such as removing copyrighted data from LLMs (Eldan & Russinovich, 2023), as well as practical necessity of purging outdated or incorrect information (Dhingra et al., 2022). As models scale in size and training cost, understanding unlearning methods is becoming an important research frontier in trustworthy and adaptive NLP systems.

Concurrently, the phenomenon of *mode connectivity* in deep learning has shown that independently trained models can often be connected by low-loss paths in parameter space (Garipov et al., 2018; Qin et al., 2022), as illustrated in §2, Figure 1a. These findings have important implications for understanding loss landscape, model ensembling, and generalization (Garipov et al., 2018; Zhao et al., 2020).

However, existing studies on most mode connectivity has focused largely on image classification tasks (Draxler et al., 2018; Vrabel et al., 2025), with straight-forward optimization objectives and static data distributions. Its relevance to unlearning remains unexplored. In addition, understanding of underlying loss landscape of unlearning remains largely unexplored, despite recent advances (Liu et al., 2024b;c; Hong et al., 2024b). In particular, it is unclear whether mode connectivity holds during unlearning, and what this reveals about unlearning.

This paper introduces and formalizes the concept of **Mode Connectivity in Unlearning (MCU)**–a framework to analyze the loss landscape during unlearning and to assess whether different unlearning strategies converge to mechanistically similar solutions. Specifically, we investigate the following research questions: **RQ1**: what does the loss landscape of unlearning look like under various training conditions (e.g., curriculum learning, second-order optimization)? **RQ2**: can MCU reveal understandings of unlearning methods, such as mechanistic similarity or differences between different methods?

Answering these questions provides insight into the generalization, stability, and interpretability of unlearning methods. For instance, the existence of a smooth and low-loss path between two

unlearning solutions may indicate shared inductive biases or similar optimization dynamics, which suggest that the unlearning methods reside in connected regions of the loss landscape. Conversely, a lack of connectivity may indicate divergent training behaviors and different solution structures.

Through extensive experiments on diverse tasks and different training paradigms, we find that the emergence of MCU is highly influenced by factors such as the choice of unlearning method, the complexity of unlearning task, and data marked for unlearning. We find that unlearned models trained with fundamentally different optimization techniques can converge to the same low-loss manifold. In addition, although the same manifold can yield models with similar losses, their performances on different evaluation metrics can vary significantly (§ 5). We further investigate how different unlearning methods are mechanistically (dis)similar using linear mode connectivity. Finally, we discuss how mode connectivity can improve the effectiveness of unlearning, as well as defending against adversarial attacks (§ 6). Our contributions are:

- **Mode Connectivity in Unlearning (MCU)**: We introduce and formalize MCU as a novel framework for studying machine unlearning through the lens of loss landscape (§3). To the best of our knowledge, this is the first study of mode connectivity in the unlearning setting.

- **Novel Experimental Conditions for Mode Connectivity**: We examine MCU under a range of new experimental conditions in mode connectivity, including curriculum learning, second-order optimization, and unlearning methods. These conditions have not previously been investigated in mode connectivity literature.

  These experiments provide new empirical insights into how optimization techniques influence unlearning (§3.1).

- **Insights into Unlearning**: We reveal mechanistic similarities, generalization, unlearning difficulty with MCU. These insights provide new directions for the development of robust unlearning methods (§6).

## 2 PRELIMINARIES

**Notation**  Let $f_{\theta_o}$ be a model trained on dataset $D$ with task loss $L$. In addition, assume that $D$ can be divided into two disjoint sets: the forget set $D_f$ and the retain set $D_r = D \setminus D_f$.

**Machine Unlearning**  Machine unlearning aims to remove the influence of the forget set $D_f$ from the trained model $f_{\theta_o}$ and preserve the knowledge of retain set $D_r$. A good unlearning model $f'$ should achieve high loss on $D_f$ and low loss on $D_r$. A commonly used solution is to fine-tune the original model $f_{\theta_o}$ to minimize the task loss on $D_r$ while maximizing the task loss on $D_f$ (Jia et al., 2024b; Cheng & Amiri, 2024). For example, GradDiff (Maini et al., 2024) directly implements the above approach:

$$f' = \arg\min_{\theta'} L(D_r) - L(D_f). \tag{1}$$

Details of related work and additional unlearning methods are discussed in Appendix A and B.

**Mode Connectivity**  Let $\theta_1$ and $\theta_2$ denote the weights of two independently trained models on some dataset $D$ using loss $L$. The objective of mode connectivity is to find a curve $\phi(t) \to \mathbb{R}^{|\theta|}, t \in [0, 1]$ in the parameter space that connects the two minimizers $\theta_1$ and $\theta_2$, where $\phi(0) = \theta_1$ and $\phi(1) = \theta_2$. Curve $\phi(t)$ connecting $\theta_1$ and $\theta_2$ satisfies mode connectivity if the path $\phi(t)$ does not yield "barriers," defined as sudden increase in loss (Garipov et al., 2018; Lubana et al., 2023). Formally, $\forall t \in [0, 1]$:

$$L\big(D; \phi(t)\big) \leq t \cdot L(D; \theta_1) + (1 - t) \cdot L(D; \theta_2). \tag{2}$$

In the loss landscape, mode connectivity tries to find a low loss path $\phi$ connecting $\theta_1$ and $\theta_2$ without hitting any barrier. In other words, every set of parameter induced by $\phi(t)$ yield comparable performance to the minimizers $\theta_1$ and $\theta_2$. The parametrization of $\phi$ determines the shape of the curve connecting the two minimizers $\theta_1, \theta_2$. Below, we present two commonly used curve types:

- **Linear:** a linear interpolation of minimizers with no optimization involved, i.e. $\phi(t) = t\theta_1 + (1 - t)\theta_2$. Stronger linear connectivity indicates stronger mechanistic similarity of minimizers, such as their inductive biases (Lubana et al., 2023).

Figure 1: **(a)**: Illustration of standard mode connectivity (MC): MC finds a smooth curve connecting two unlearners that yields consistent low loss on $D$. **(b)**: Illustration of mode connectivity in unlearning (MCU): unlearning removes knowledge of forget set $D_f$ from the trained model $f_{\theta_o}$ while maintaining knowledge of retain set $D_r = D \setminus D_f$. MCU finds a smooth curve connecting the two unlearned models $\theta'_1$ and $\theta'_2$ that yields consistent low loss on $D_r$ and high loss on $D_f$. See details in § 3.

- **Quadratic:** a smooth quadratic curve connecting two minimizers, i.e $\phi(t) = (1-t)^2\theta_1 + 2t(1-t)\theta_{12} + t^2\theta_2$, where $\theta_{12}$ needs to be trained explicitly.

To find non-linear curve, we can minimize the total accumulated loss along the curve to find the midpoint $\theta_{12}$ which will later be used for interpolation. More details of the curve finding optimization are discussed in Appendix D.

## 3 MODE CONNECTIVITY IN UNLEARNING (MCU)

**Definition 1** (Mode Connectivity in Unlearning (MCU)). *As illustrated in Figure 1, let $\theta'_1$ and $\theta'_2$ denote the weights of two unlearned models of applying some unlearning procedure $\mathcal{U}$ on the original model $f_{\theta_o}$ with different configurations. MCU holds if there exists a path $\phi_{\theta'_1 \to \theta'_2}(t)$ in parameter space that connects $\theta'_1$ and $\theta'_2$ without yielding barriers. Formally, $\forall t \in [0, 1]$*

$$L\big(D_r; \phi(t)\big) \leq t \cdot L(D_r; \theta'_1) + (1-t) \cdot L(D_r; \theta'_2), \tag{3}$$

$$L\big(D_f; \phi(t)\big) \geq t \cdot L(D_f; \theta'_1) + (1-t) \cdot L(D_f; \theta'_2). \tag{4}$$

In MCU, "barriers" on the retain set $D_r$ refer to the sudden *increase* of task loss $L$ (as in standard mode connectivity), while "barriers" on the unlearn set $D_f$ refer to the sudden *decrease* of task loss. Eq. 3 ensures that the task loss on the retain set $D_r$ remains both low and smooth along the mode connectivity curve, indicating consistent model behavior during the unlearning process. Similarly, Eq. 4 enforces a high and smooth loss on the forget set $D_f$ along the mode connectivity path. In other words, MCU is realized when there exists a continues path of model weights connecting the unlearners $\theta'_1$ and $\theta'_2$, such that performance remains high on $D_r$ and low on $D_f$ along the curve.

**Connection to Standard Mode Connectivity** In contrast to standard mode connectivity, MCU must satisfy objectives on both $D_f$ and $D_r$. Essentially, MCU examines whether it is possible to find a continues curve between two unlearners such that there are no significant loss barriers in **two distinct loss landscapes**–a key difference compared to standard mode connectivity, which typically considers only a single task or dataset. Another key difference is that in standard mode connectivity (Garipov et al., 2018), the unlearners $\theta_1$ and $\theta_2$ are obtained by training the model from two random initializations. In contrast, the unlearners $\theta'_1$ and $\theta'_2$ in MCU are both derived from the exact same trained model $f_{\theta_o}$.

### 3.1 INFLUENCE OF TRAINING DYNAMICS ON MODE CONNECTIVITY IN UNLEARNING

Unlearned models can be trained by various paradigms, such as curriculum learning (CL) (Barbulescu & Triantafillou, 2024; Zhao et al., 2024; Cheng & Amiri, 2024) and second-order (SO) optimization (Jia et al., 2024b; Zhang et al., 2024a). To systematically analyze the impact of these factors, we define several novel experimental conditions and evaluate MCU under these conditions.

**Robustness Against Randomness** We begin with a classical mode connectivity setup: training two unlearners (unlearned models in our case) independently from different random seeds. This examines whether MCU is robust to stochasticity in optimization.

Table 1: Experimental settings to study MCU. Rand: unlearners trained with different random seeds, CL and SO: curriculum learning and second-order optimization respectively, Met: unlearners trained with different unlearning methods. All settings except for "Rand" are novel in mode connectivity.

| unlearners unlearned with | Standard | Both CL | Both SO | Mixed CL/Non-CL | Mixed FO/SO |
|---|---|---|---|---|---|
| Different randomness | Rand | Rand-CL | Rand-SO | CL-Non-CL | FO-SO |
| Different unlearning methods | Met | Met-CL | Met-SO | Met-CL-Non-CL | Met-FO-SO |

**Sensitivity to Training Curricula**   Curriculum learning (CL) (Bengio et al., 2009), where training data is introduced to a learner in a specific order, contributes to the efficacy of unlearning (Zhao et al., 2024; Cheng & Amiri, 2024). We investigate two CL-based settings: (a): both unlearners are trained with CL (**-CL**); and (b): one unlearner is obtained through CL and the other does not (**CL-Non-CL**). These configurations allow us to assess whether changes in sample learning order affect the emergence of MCU.

**Connectivity across Optimization Orders**   While most unlearning methods use first-order (FO) gradients, recent studies demonstrate that unlearning can benefit from second-order information (SO) (curvature via Hessian) in case of LLMs (Jia et al., 2024b) and discriminative tasks (Cheng & Amiri, 2024). We analyze MCU under two different SO settings: (a): both unlearners are trained with SO (**-SO**); and (b): one unlearner is trained with FO and the other trained with SO (**FO-SO**). This analysis helps us understand how different optimization dynamics shape the unlearning loss landscape.

**Similarity across Unlearning Methods**   All unlearning methods share the common objective of removing knowledge of $D_f$ while retaining knowledge of $D_r$. We examine whether unlearners derived from different unlearning methods can be smoothly connected. We hypothesize that methods with similar formulation and inner mechanisms are more likely to establish connectivity. This experiment provides a lens into mechanistic similarity of different unlearning algorithms (Lubana et al., 2023).

**Experimental Novelty**   To the best of our knowledge, with the exception of the randomness factor (see "Robustness Against Randomness" above), all other factors discussed above are novel within the mode connectivity literature. Multiple configurations can be combined, as summarized in Table 1. Together, they provide diverse and realistic perspectives on the training dynamics of unlearning, broaden the scope of mode connectivity research, and deepen our understanding of the factors that enable or prevent successful unlearning.

## 4 EXPERIMENTAL SETUP

**Datasets and Forget Sets**   We analyze MCU on widely adopted LLM unlearning and classification unlearning benchmarks. For LLM unlearning, we use TOFU (Maini et al., 2024), MUSE (Shi et al., 2025), and WMDP (Li et al., 2024) dataset. For classification, we use three datasets from MU-Bench (Cheng & Amiri, 2024): image classification on CIFAR-10 (Krizhevsky, 2009), biomedical text relation classification on DDI2013 (Segura-Bedmar et al., 2013), and image-text visual entailment on NLVR2 (Maas et al., 2011). The original models and standard data splits are provided by the above benchmarks. Specifically, TOFU has 1%, 5%, 10% of forget set. MUSE has `Books` and `News` as forget sets, while WMDP has `Cyber-security` (Cyber) and `Bio-weapons` (Bio) as forget sets. MU-Bench provides 2%, 4%, 6%, 8%, 10% forget set.

**Unlearning Methods**   We use MCU to analyze the following LLM unlearning methods: 1) Gradient Ascent (GA) (Golatkar et al., 2020), 2) GradDiff (GD) (Maini et al., 2024), 3) Negative Preference Optimization (NPO) (Zhang et al., 2024b), 4) SimNPO (Fan et al., 2024b), 5) RMU (Li et al., 2024), and 6) WAGLE (WGA) (Jia et al., 2024a). For classification tasks we use: 1) Gradient Ascent (GA) (Golatkar et al., 2020), 2) Random Labeling (RL) (Graves et al., 2021), 3) Bad Teaching (BT) (Chundawat et al., 2023), and 4) Saliency Unlearning (SU) (Fan et al., 2024c). These methods cover a diverse set of unlearning paradigms and are commonly used in existing works. Appendix B provides additional details.

**Evaluation**   To evaluate MCU, we sample multiple points by varying the interpolation weight $t \in [0, 1]$ with small step size. Each value of $t$ induces a set of model weights according to the

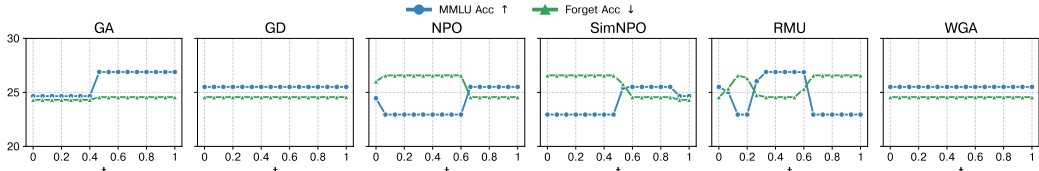

Figure 2: MCU under **Rand** setting on **WMDP Cybersecurity dataset**. Additional results are shown in Appendix E Figure 7–21.

parametrization of the curve $\phi_\theta(t)$ (§ 2). Following previous mode connectivity work on language models (Qin et al., 2022), we sample 16 points with equal step size in $[0, 1]$.

For each induced unlearned model, we use standard metrics from each benchmark. On classification unlearning tasks, we use accuracy on test set $D_t(\uparrow)$, accuracy on forget set $D_f(\downarrow)$, accuracy on retain set $D_r(\uparrow)$, and Zero-Retrain Forgetting score ZFR ($\uparrow$) which measures the prediction similarity of $D_f$ between unlearned and original models. On TOFU, we use Forget Quality ($\uparrow$) and Model Utility ($\uparrow$) which is a $p$-value of Kolmogorov-Smirnov test (KS-Test). On MUSE, we use forget_verbmem ($\downarrow$), forget_knowmem ($\downarrow$), retain_knowmem ($\downarrow$), extraction strength ($\downarrow$), and privacy leakage (privleak, $\downarrow$). On WMDP, we use accuracy on forget set ($\downarrow$) and accuracy on MMLU evaluation set ($\uparrow$). Appendix C provides additional details and metrics.

## 5 RQ1: LOSS LANCSCAPE ANALYSIS OF MACHINE UNLEARNING

We investigate the conditions under which mode connectivity in unlearning (MCU) emerges across different models, datasets, and optimization strategies. Our results show that MCU is not only possible but often prevalent–though, its emergence is influenced by unlearning method, training dynamics and the size of forget set $D_f$.

### 5.1 MCU ACROSS DIFFERENT CONDITIONS

On WMDP Cybersecurity, we observe almost perfectly smooth curve with no degradation of unlearning quality with GD and WGA. These curves show consistently high forget quality (low forget accuracy) and retained model utility (high MMLU accuracy), which suggests that unlearning solutions reside on a connected low-loss manifold. This observation aligns with findings in standard mode connectivity (Draxler et al., 2018), where minima are not isolated but from a single connected manifold of low loss in parameter space. For RMU, we observe significant fluctuations in both forget quality and retained model utility, particularly in the middle part. Similarly, there are ridges on the loss landscapes of NPO and SimNPO but less significant. The existence of mode connectivity paths suggests that modern neural networks have enough parameters such that they can achieve good predictions while a big part of the network undergoes structural changes.

On classification dataset, results vary. GA results in the smoothest MCU curves, both linear and quadratic, particularly for small $|D_f| = 1\%$. Due to the similarity in design, RL and SU have very similar MCU patterns. Both types of curve yield models with degraded forget set performance ($\downarrow$) in the middle part of the curve (green line in Figure 17), with more prominent degradation on linear than quadratic curves. On BT, there is a strong linear MCU but the curve finding process fails to converge to meaningful quadratic MCU. This suggests that simpler connectivity may appear but hard to detect. We hypothesize that BT has a more rugged loss landscape than other methods, possibly because of its indirect loss formulation based on representations rather than explicit tasks loss. These results highlight the difference in loss landscape of unlearning methods. More details are discussed in Appendix E.1.

**Smooth local optima** In most cases, we observe that smooth manifold appear when both minimizers achieve low loss, which is consistent with early works on mode connectivity (Draxler et al., 2018; Garipov et al., 2018). However, in unlearning, worse performing minimizers do not necessarily mean smoother manifold in loss landscape. One example is the comparison between Rand (Figure 7) and Rand-CL (Figure 8) when unlearning with RL on TOFU dataset. While on RL, CL yield minimizers with slightly lower forget quality when $|D_f = 1\%|$, $|D_f = 5\%|$, and equally low forget quality when $|D_f = 10\%|$. However, worse performing minimizers do not necessarily mean smoother manifold in

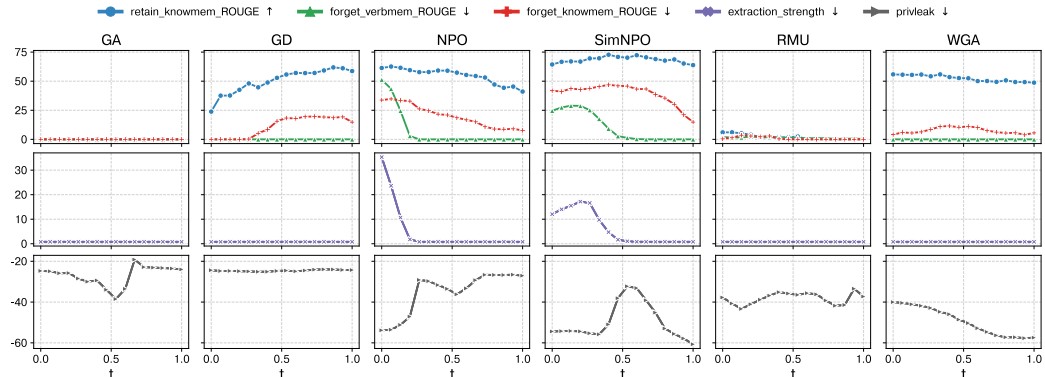

Figure 3: MCU under **CL-Non-CL** setting on **MUSE Books dataset**.

loss landscape. Specifically, we find that RL yields a mode connectivity curve with less fluctuations, i.e. less variation in forget quality, on both linear and quadratic curve.

**Finding MCU Can Be Difficult**   Quadratic MCU involves finding the midpoint of the curve, an unlearned model using the same unlearning method as $\theta'_1$ and $\theta'_2$. In some cases, this can be hard to optimize. For example, linear MCU exists while quadratic MCU yields much worse curves, see Figure 17 BT method. This is because quadratic MCU requires the underlying unlearning method to find the curve, where some methods fall short during optimization. We observe this phenomenon on BT most frequently, which indicates that BT suffer from significant computation cost and low convergence speed.

Experiments also show that unlearning effectiveness does not guarantee smooth mode connectivity. SalUn is generally an effective unlearning method on classification tasks. However, it often fails to yield smooth mode connectivity curves, see Figure 17–19. This is potentially due to the sparse nature of SalUn. This suggests that strong unlearning efficacy does not necessarily imply the existence of smooth paths in the loss landscape.

## 5.2   EFFECT OF TRAINING PARADIGMS

**Both Curriculum Learning (CL)**   When both endpoints (minimizers) are unlearned with CL (Rand-CL), we observe different contributions from CL across different datasets. On TOFU, GA, GD, or NPO result in connectivity patterns that are equally as performant as Non-CL minimizers, see Figures 8 and 7). This is while RL yields minimizers with slightly lower forget quality but smoother mode connectivity curves, for both linear and quadratic. This suggests that CL-based unlearning may converge to different regions than non-CL-based unlearning, resulting in comparable and sometimes-better performance. On classification tasks, finding MCU in CL space (Figure 18) is much easier than non-CL space (Figure 17) for BT. However, CL has trivial contribution on RL and SU. These results imply that CL can guide optimization toward flatter regions of the loss landscape, depending on the model and method.

**Both Second-Order Optimization (SO)**   Similar to CL, when both minimizers are unlearned with second-order optimization (SO), we observe different effects incurred by SO on generative and discriminative tasks, and on different size of forget set $|D_f|$. On TOFU, So-based unlearning results in more pronounced barriers (Rand-SO in Figure 9) compared to standard unlearning (Rand in Figure 7) and CL-based unlearning (Rand-CL in Figure 7). These results can be attributed to SO optimization, which takes larger steps in parameter space than FO optimization (Liu et al., 2024a). This may sometimes lead to different low-loss manifolds. On classification datasets, SO generally leads to a smoother manifold for all methods, where linear and quadratic connectivity are easier to emerge for all methods, even when $|D_f|$ is large. Still, SO is insufficient for RL and SU when unlearning large forget sets, e.g. $|D_f| \geq 8\%$.

**CL and Non-CL**   MCU may emerge between models trained with and without CL. On MUSE, RMU and WGA show relatively smooth connectivity, while other methods show large ridges, see Figure 3. This suggests that CL-based unlearning can sometimes converge to a different manifold than standard unlearning, yielding a more effective or less effective unlearner.

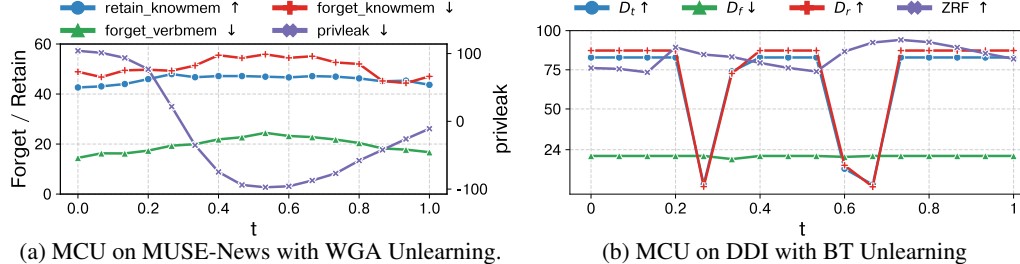

(a) MCU on MUSE-News with WGA Unlearning.  (b) MCU on DDI with BT Unlearning

Figure 4: Same MCU curve can have both smooth and fragmented loss landscapes measured by different losses.

**FO and SO**   MCU across FO and SO minimizers show more diverse patterns. On TOFU, Figure 11, GD but not other unlearning methods show smooth MCU. Similarly, on classification datasets (Figure 21), smooth MCU hardly emerges. These results suggest that FO and SO optimization can drive the same unlearning method converge to different low-loss manifold. These minima are not connected by smooth pathways, demonstrated by failure of quadratic MCU. There is no consistent results that FO or SO is better.

## 5.3    FRAGMENTATION OF LOSS LANDSCAPE AND DIVERGENT PERFORMANCE

**Connectivity differs on $D_r$ and $D_f$**   On retain set $D_r$, smooth connectivity are more likely to occur and easier to find. On the other hand, the loss landscape over the forget set can be highly irregular and prone to fluctuations. This could be because $D_r$ is typically much larger than $D_f$, which leads to a more stable optimization signal and a smoother curve in the retain region of the loss landscape.

**Similar loss divergent performance**   While smooth MCU confirms that two unlearners lie in the connected low-loss regions, this does not always translate to similar performance acorss all evaluation metrics. In other words, loss landscape may be fragmented despite smooth loss on $D_f$.

For example, with WGA on MUSE-News, we observe a relevatively smooth MCU curve, with trivial performance variation on forget_verbmem, forget_knowmem, and retain_knowmem. However, performance on privleak can fluctuate significantly along the same MCU curve, ranging from +100 to -100, see Figure 4(a). This indicates that although close to each other in parameter space, some unlearners are extremely prone to attackers and may leak information, while others may be much more robust.

Another example is with BT on DDI. We observe stable accuracy on $D_f$ along the MCU curve, but significant fluctuations on $D_r$, $D_t$ and moderate fluctuation on ZRF, see Figure 4(b). This indicates that that although close to each other in parameter space, some unlearners maintain good amount of knowledge from the original model ($D_r$ and $D_t$), while others may have completely forgotten.

This behavior demonstrates that although many unlearners may reside in the same minima with similar loss, their unlearning behavior can differ significantly. Same curve can have divergent behavior under different loss landscapes. This also shows a limitation of current evaluation protocols for unlearning and motivates the need for richer and intrinsic evaluation on model parameters (Hong et al., 2024a).

## 6    RQ2: UNDERSTANDING MACHINE UNLEARNING WITH MCU

MCU can indicate several characteristics of unlearning, including 1) mechanistic similarity between unlearning methods, 2) generalization, 3) robustness to attacks, and 4) unlearning difficulty.

### 6.1    MCU INDICATES MECHANISTIC (DIS)SIMILARITY BETWEEN UNLEARNING METHODS

Prior works have found that linear mode connectivity indicates if minimizers share internal mechanisms for making predictions (Lubana et al., 2023). We use this to analyze the similarity between two unlearning methods.

Figure 5 shows the pair-wise linear MCU between all methods. We can see that GA shows smooth MCU on forget set (knowmem and verbmem) with all other unlearning methods, indicating that all methods share similar internal mechanisms of handling forget set samples. They do differ in retain set significantly (Row 1 $t \rightarrow 1$), since GA does not have retain mechanism.

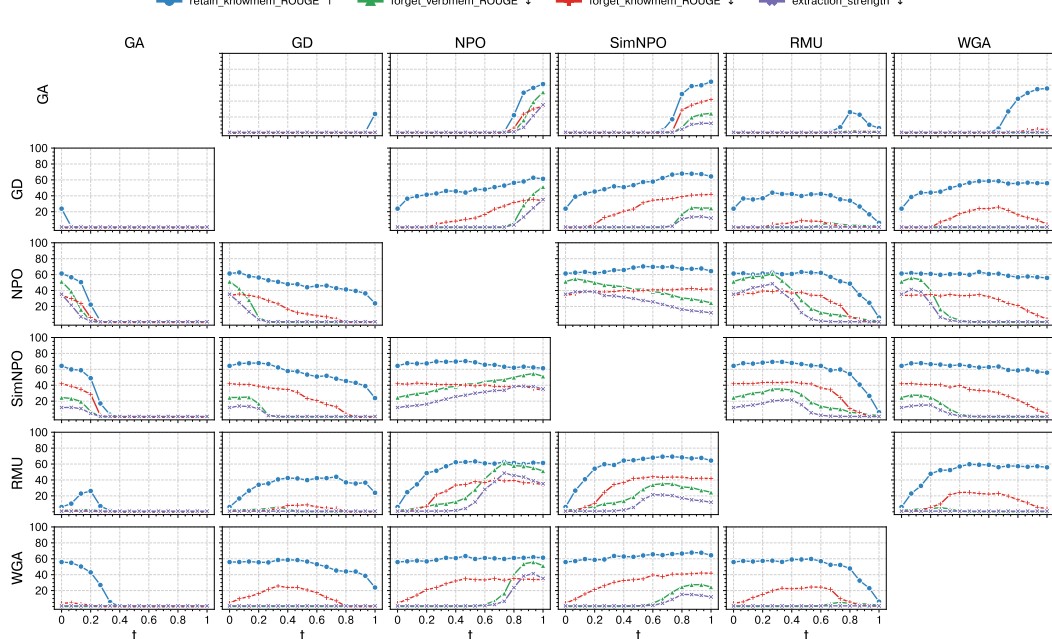

Figure 5: MCU under **Met** setting on **MUSE Books dataset**. Methods on rows and columns correspond to $\theta_1'$ and $\theta_2'$ respectively. MCU is symmetric. Additional results are shown in Appendix E, Figures 12–16, Figures 22–26.

We observe high similarity on retain knowledge between NPO and SimNPO, an improved version of NPO. However, they differ from each other on forget knowledge and extraction strengths. Both of them are significantly different from GD, since GD simply minimizes loss on $D_r$ and maximizes loss on $D_f$, while NPO and SimNPO involves implicit reward modeling.

RMU and WGA both differ from all other methods. This is because WGA only trains sparse parameters. RMU also updates a subset of parameters but a few dense layers. Moreover, it incorporates matching activation patterns as an extra loss.

## 6.2 MCU INDICATES GENERALIZATION

Prior work has found that mode connectivity indicates generalization, where minimizers converge to a minima with smooth loss landscape and constant low error, potentially leading to improved performance (Garipov et al., 2018; Wang et al., 2023).

Table 2: MCU improves generalization.

| Acc (↓) | NPO | + MCU Ensemble |
|---------|------|----------------|
| 63.7    | 24.1 | 21.5           |

Previous experiments demonstrate that intermediate unlearners along the MCU curve may outperform both endpoints. We then propose a generalization method, sampling $N = 3$ unlearners along the curve with equal distance. We average the parameters of the sampled models and two endpoints, resulting in a new merged unlearner. Results on WMDP show that our ensemble strategy can outperform the endpoints in both unlearning effectiveness (forget accuracy), see Table 2. This suggests that interpolated models may achieve a better trade-off between forgetting and retaining, and presents a promising directions for ensembling or model selection using mode connectivity. Other examples include RL on NLVR2.

## 6.3 MCU MAKES UNLEARNING ROBUSTNESS TO ATTACKS

It is well established that a smoother loss landscape is associated with greater robustness in deep learning models (Zhang et al., 2017; Foret et al., 2021; Zhang et al., 2024c; Li et al., 2025). Specifically, minimizers along smooth mode connectivity curves are typically more robust than those obtained by standard training. Building on this insight, our findings suggest that MCU can identify unlearned models that are more resilient to adversarial threats, such as backdoor attacks (Lin et al., 2023) and relearning attacks (Hu et al., 2024).

Following prior work on unlearning robustness (Fan et al., 2025), we first apply the unlearning method to models trained on the WMDP Cybersecurity benchmark. We then apply a relearning attack by fine-tuning the unlearned model on generic Wikipedia text and subsequently evaluate its unlearning effectiveness post-attack.

As shown in Figure 6, models unlearned via standard NPO methods are susceptible to relearning, with their forgetting effects quickly deteriorating. In contrast, models obtained through MCU—by averaging across multiple unlearners—demonstrate markedly greater robustness to such attacks. We attribute this improvement to the smoother loss landscape induced by the averaging process.

This observation is consistent with recent work showing that minimizing loss landscape sharpness (encouraging smoothness) during optimization enhances the robustness to adversarial attacks for LLM unlearning (Fan et al., 2025).

### 6.4 MCU INDICATES UNLEARNING COMPLEXITY

Prior unlearning works identify challenging forget sets using bi-level optimization (Fan et al., 2024a), proximity to test set (Cheng et al., 2023), and LLM-generated stress-test set (Cheng & Amiri, 2025). We propose that the smoothness of MCU can be an additional indicator of the difficulty of unlearning task, grounded in the geometric properties of the loss landscape. The smoothness of MCU can reflect the sharpness of the loss landscape. A highly irregular loss landscape may indicate high difficulty of optimization, and balance the forget-retain tradeoff.

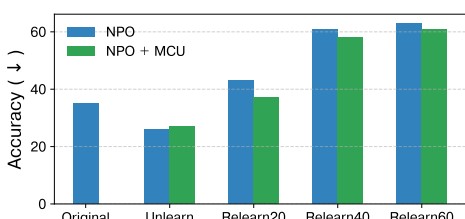

Figure 6: MCU finds unlearners with smooth loss landscape and robust to relearning attack.

We observe that MCU appears more often on simpler tasks such as WMDP, TOFU and DDI, where even methods like GA–which typically over-forget–show smooth connectivity. This aligns with parallel findings that LLM unlearning benchmarks does not show progress of unlearning (Thaker et al., 2025). Another potential reason is that the $p$-value-based evaluation is not sufficient to reflect the true quality of unlearning methods, which aligns with prior findings (Liu et al., 2024b).

In contrast, MUSE has a more disconnected loss landscape with steep transitions, which suggest higher difficulty than WMDP. These patterns align with prior findings that link unlearning difficulty to factors like sample memorization (Barbulescu & Triantafillou, 2024) and interdependence of forget-retain sets (Zhao et al., 2024), which suggest tasks complexity and data structure considerably shape the loss landscape.

## 7 CONCLUSION

We introduce and formalize mode connectivity in unlearning (MCU) as a framework for understanding the loss landscape and optimization dynamics of machine unlearning. To the best of our knowledge, this is the first work that studies the loss landscape of unlearning with mode connectivity in various conditions. We find that the emergence of mode connectivity can be influenced by task complexity, forget set size, and optimization strategies like curriculum learning and second-order methods.

Our experiments across diverse tasks, unlearning methods, and training configurations show that MCU provides insights into smoothness of similarity, training stability, and unlearning difficulty. We show that MCU can be used as a diagnostic tool and open new directions for improving unlearning methods. In future, we plan to use mode connectivity to identify intermediate models along unlearning paths that optimize the trade-off between forgetting and retaining, and improve robustness of unlearning (Jung et al., 2024; Huang et al., 2025).

**Limitations** In this study, we focus on parameters-space mode connectivity, whereas recent work discovers input-space mode connectivity (Vrabel et al., 2025). In addition, we did not investigate multi-dimensional manifolds(Benton et al., 2021) in mode connectivity and examine alternative curve types such as star-shaped and geodesic mode connectivity(Lin et al., 2024).

## ETHICS STATEMENT

This work focuses on improving the transparency and reliability of machine unlearning, which is motivated by ethical considerations such as user data privacy, regulatory compliance, and the right to be forgotten. All experiments are conducted on publicly available datasets, and no personally identifiable or sensitive data is used.

## REPRODUCIBILITY STATEMENT

Our experiments are based on public implementations from the benchmarks we used. We will release our code upon acceptance.

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

## A  RELATED WORK

**Machine Unlearning**  Early unlearning methods span across efficiently retraining (Bourtoule et al., 2021b; Wu et al., 2020), model pruning (Jia et al., 2023), manipulating gradients (Ullah et al., 2021; Hoang et al., 2024), adversarial unlearning (Setlur et al., 2022; Wei et al., 2023), and data augmentation (Choi et al., 2024). Unlearning on LLMs recently draws more attention (Eldan & Russinovich, 2023; Ji et al., 2024; Kassem et al., 2023; Cheng & Amiri, 2025). However, there is less attention on mechanistically understanding the loss landscape of machine unlearning methods.

**Mode Connectivity**  Furthermore, several studies have shown that independently trained minimizers can be connected by low loss paths, a phenomenon known as mode connectivity (Draxler et al., 2018; Garipov et al., 2018; Frankle et al., 2020), across both vision and language models (Qin et al., 2022). During pruning, linear mode connectivity emerges only at early stage of training. This connectivity has been extended to multi-dimensional manifolds(Benton et al., 2021), and alternative topologies such as star-shaped and geodesic connectivity(Lin et al., 2024). Mode connectivity can also lead to more effective (Garipov et al., 2018) or adversarially robust (Zhao et al., 2020; Wang et al., 2023) models if ensembling along the curve. Vrabel et al. (2025) discover that mode connectivity can happen in input space. Existing works on mode connectivity focus on the learning process. There is no prior work that investigates mode connectivity in machine unlearning.

## B  DETAILS OF UNLEARNING METHODS

Below, we present the details of the unlearning methods used in our study.

**Gradient Ascent**  Gradient Ascent (GA) (Golatkar et al., 2020) performs gradient ascent on $D_f$ without any mechanism to maintain utility on the retain set $D_r$.

**Random Labeling**  Random Labeling (RL) (Golatkar et al., 2020) fine-tunes $f_{\theta_o}$ on $D_f$ with corrupted labels and the original $D_r$ (or a fraction of it if the entire $D_r$ is too large). This method aims to inject errors to the forget set.

**Saliency Unlearning**  SalUn (SU) (Fan et al., 2024c) first finds parameter that are salient to unlearning $D_f$. Next, it performs Random Labeling but only updates the salient parameters.

**Bad Teaching**  Bad Teaching (BT) (Chundawat et al., 2023) forces the unlearned model to predict $D_r$ similarly to the original model and to predict $D_f$ similarly to an incompetent model (e.g. a randomly initialized model). It minimizes the KL-Divergence between prediction logits $\mathbb{KL}(f'(D_r)||f(D_r))$ on $D_r$ and maximizes KL-Divergence between prediction logits $\mathbb{KL}(f'(D_f)||f_d(D_f))$ on $D_f$, where $f_d$ is the incompetent model, e.g. a randomly initialized model.

**Gradient Difference**  GradDiff (GD) (Maini et al., 2024) minimizes task loss on $D_r$ and maximizes task loss on $D_f$.

**Negative Preference Optimization**  NPO (Zhang et al., 2024b) is built upon the DPO (Rafailov et al., 2023) algorithm to post-train LLMs. In the original DPO, each query $q$ corresponds to a winning response $y_w$ to prioritize and a losing response $y_l$ to suppress. NPO functions only the losing response with no winning response.

## C  DETAILS OF EVALUATION METRICS

We provide detailed descriptions of the evaluation metrics used in our analysis. On MU-Bench (Cheng & Amiri, 2024) tasks, we follow the original paper to adopt accuracy as the evaluation metric. In addition, we employ Zero-Retrain Forgetting score (↑) (Chundawat et al., 2023), which measures the similarity of prediction logits on $D_f$ between the unlearned model and a random model.

TOFU (Maini et al., 2024) evaluates the unlearned model using $p$-value of Kolmogorov-Smirnov test for Model Utility (↑) and Forget Utility (↑), which measure the similarity of probability distributions

between the unlearned and retrained model. Additionally, we also include verbatim evaluation using ROUGE-L recall score on Retain Authors ($\uparrow$), Forget Authors ($\downarrow$), Real Authors ($\uparrow$), and World Knowledge ($\uparrow$).

MUSE (Shi et al., 2025) evaluates the unlearned model using verbatim memorization on forget set (forget_verbmem $\downarrow$) and knowledge memorization on forget and retain set (forget_knowmem $\downarrow$, retain_knowmem $\uparrow$), by probing the unlearned model with a series of question related to forget set. All of these scores are measured by ROUGE-L.

On WMDP (Li et al., 2024) evaluates the unlearned model using accuracy on WMDP forget set. It also evaluates the general utility of the unlearned model using the MMLU benchmark (Hendrycks et al., 2021).

## D    DETAILS OF CURVE FINDING PROCESS

To find the curve that connects $\theta_1$ and $\theta_2$, we can first compute the average loss along the curve:

$$\hat{\ell}(\theta) = \frac{\int L(\phi_\theta) \, d\phi_\theta}{\int d\phi_\theta}. \tag{5}$$

The numerator $\int L(\phi_\theta) \, d\phi_\theta$ is the line integral of the loss $L$ along the curve $\phi_\theta$. It sums up the loss values at all points along the curve, weighted by the length of the curve in the parameter space. Intuitively, it measures the total accumulated loss along the curve, accounting for how long the curve is in regions with high or low loss.

The denominator $\int d\phi_\theta$ is the total length of the curve in the parameter space. It normalizes the numerator by the total length, ensuring that the result does not depend on the specific parameterization of the curve (e.g., stretching or shrinking segments artificially).

Minimizing the above loss ensures that the path between the two sets of weights corresponds to models with consistently high accuracy.

The integrals can be rewritten in terms of the parameter $t \in [0, 1]$ as

$$\hat{\ell}(\theta) = \int_0^1 L(\phi_\theta(t)) q_\theta(t) \, dt, \tag{6}$$

$$q_\theta(t) = \frac{\|\phi_\theta'(t)\|}{\int_0^1 \|\phi_\theta'(t)\| \, dt}. \tag{7}$$

$$\mathbb{E}_{t \sim [0,1]} \hat{\ell}(\theta) = \int_0^1 L(\phi_\theta(t)) q_\theta(t) \, dt. \tag{8}$$

## E    ADDITIONAL RESULTS

We present detailed results on TOFU in Figure 7–16 and on classification datasets in Figure 17–26.

### E.1    MCU UNDER INDEPENDENTLY UNLEARNED MINIMIZERS

On TOFU, we find almost perfectly smooth curve with no degradation of unlearning quality on 3 out of 4 unlearning methods (GA, GD, and NPO). Along the curves, all model weights yield consistent unlearning quality, measured by a series of evaluation metrics, including forget quality, model utility, and ROUGE score. On the other hand when using method RL, the model weights along the curve is of consistently high quality in model utility but have slightly different forget quality. Specifically, in the middle part of the curve, we observe a drop of 0.1 point in forget quality and an increase of 0.05 point in forget ROUGE ($\downarrow$). However, since forget quality is the $p$-value of KS test, any value greater than 0.05 is considered as good unleared model, see Figure 7 for details. As the size of forget set increases, indicated by different rows in Figure 7, there is trivial variation of forget quality and model

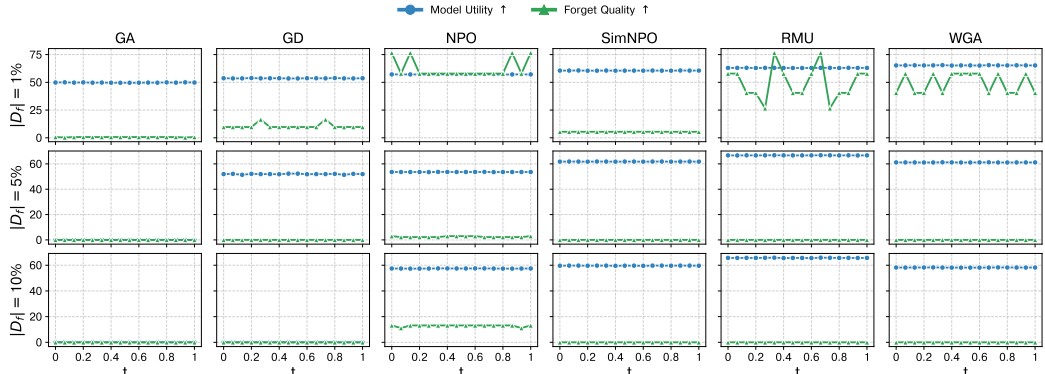

Figure 7: MCU under **Rand** setting on **TOFU dataset**.

utility along the linear and quadratic curve on GA, GD, and NPO. On RL, we notice interesting behaviors. When $|D_f| = 1\%$, forget quality degrades in the middle of the curve. When $|D_f| = 5\%$, forget quality does not change significantly. When $|D_f| = 10\%$, forget quality significantly increases in the middle of the curve. These behaviors are consistent on both linear and quadratic curves. We attribute these to the fact that RL is not an appropriate unlearning method for TOFU, which stuck in local optima and cannot ultimately converge to the low loss valley.

Therefore, we can find that the loss landscape of most unlearning methods on TOFU has essentially a flat low-loss valley where barriers, i.e. sudden performance degradation, rarely appear. This implies that, similar to learning (Draxler et al., 2018), minima of unlearning are perhaps best seen as points on a single connected manifold of low loss, rather than as the bottoms of distinct valleys for each individual unlearning method. The existence of mode connectivity paths suggests that modern neural networks have enough parameters such that they can achieve good predictions while a big part of the network undergoes structural changes. However, some unlearning methods may not converge to the low loss manifold, such as RL on TOFU dataset.

On classification dataset, we observe different patterns across different unlearning methods. On GA, it is generally easier to observe smooth MCU curve, both linear and quadratic, with small variation in forget set performance when $|D_f| = 1\%$. Due to the similarity in design, RL and SU have very similar MCU patterns. Both types of curve yield models with degraded forget set performance ($\downarrow$) in the middle part of the curve (green line in Figure 17), with more prominent degradation on linear than quadratic curves. On BT, there is a strong linear MCU but the curve finding process fails to converge to meaningful quadratic MCU. This demonstrates that simpler connectivity may appear but hard to detect. We hypothesize that BT has a more rugged loss landscape than other methods, likely because it computes loss based on representations not directly on tasks loss. These results highlight the difference in loss landscape of unlearning methods.

**CL and Non-CL**    On classification datasets, GA shows strong linear and quadratic MCU. RL and SU show quadratic (but not linear) connectivity, with slight degradation of forget set performance. This indicates that CL-based and Non-CL-based methods can converge to the same low loss manifold. On BT when $|D_f| = 4\%$, although there is almost no variation in forget set performance on linear curve, there is a major drop on retain set performance at the middle of the curve. Since MCU considers both forget set and retain set performance, this is not an emergence of MCU.

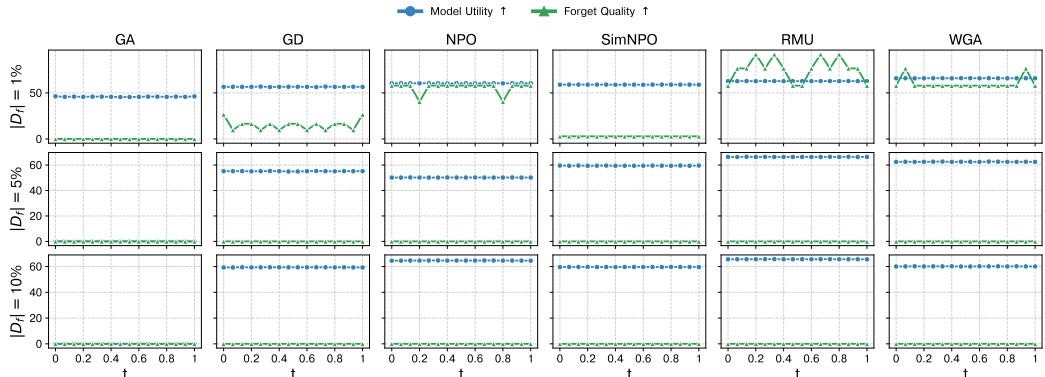

Figure 8: MCU under **Rand-CL** setting on **TOFU dataset**.

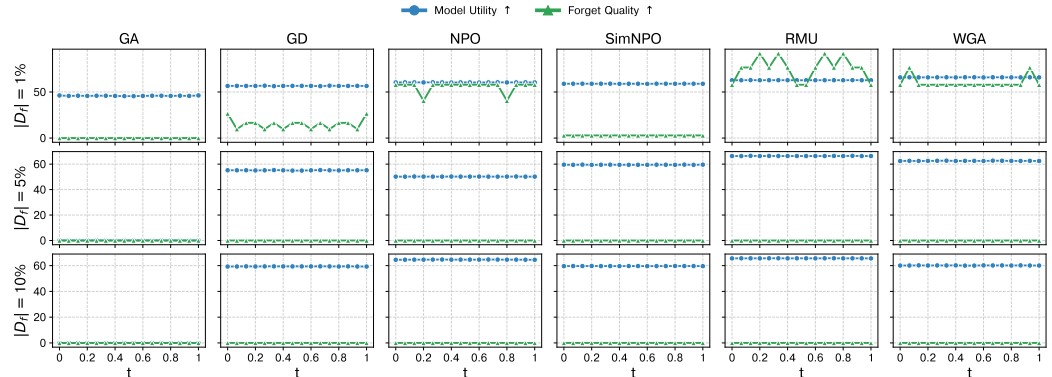

Figure 9: MCU under **Rand-SO** setting on **TOFU dataset**.

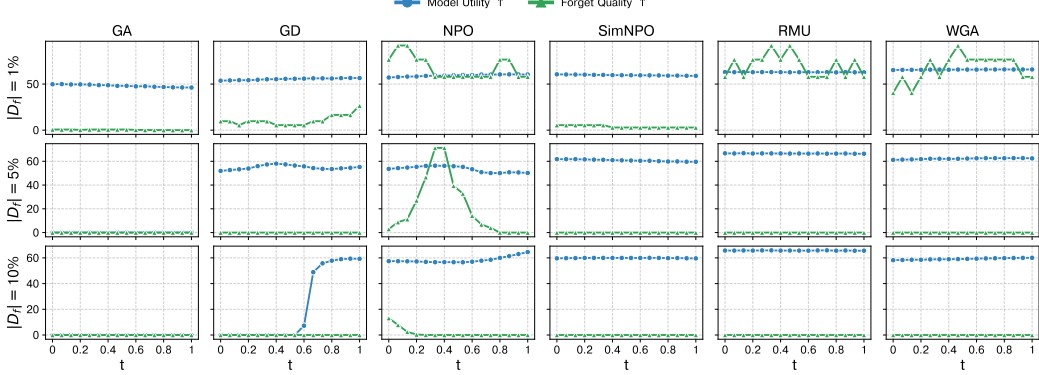

Figure 10: MCU under **CL-Non-CL** setting on **TOFU dataset**.

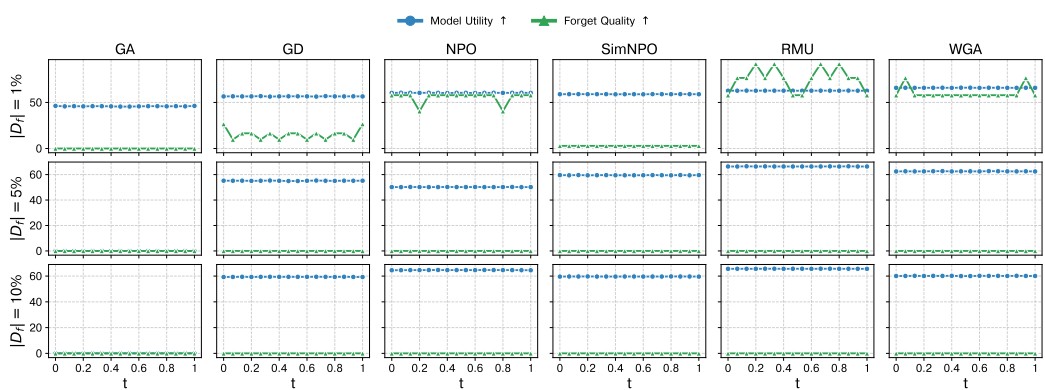

Figure 11: MCU under **FO-SO** setting on **TOFU dataset**.

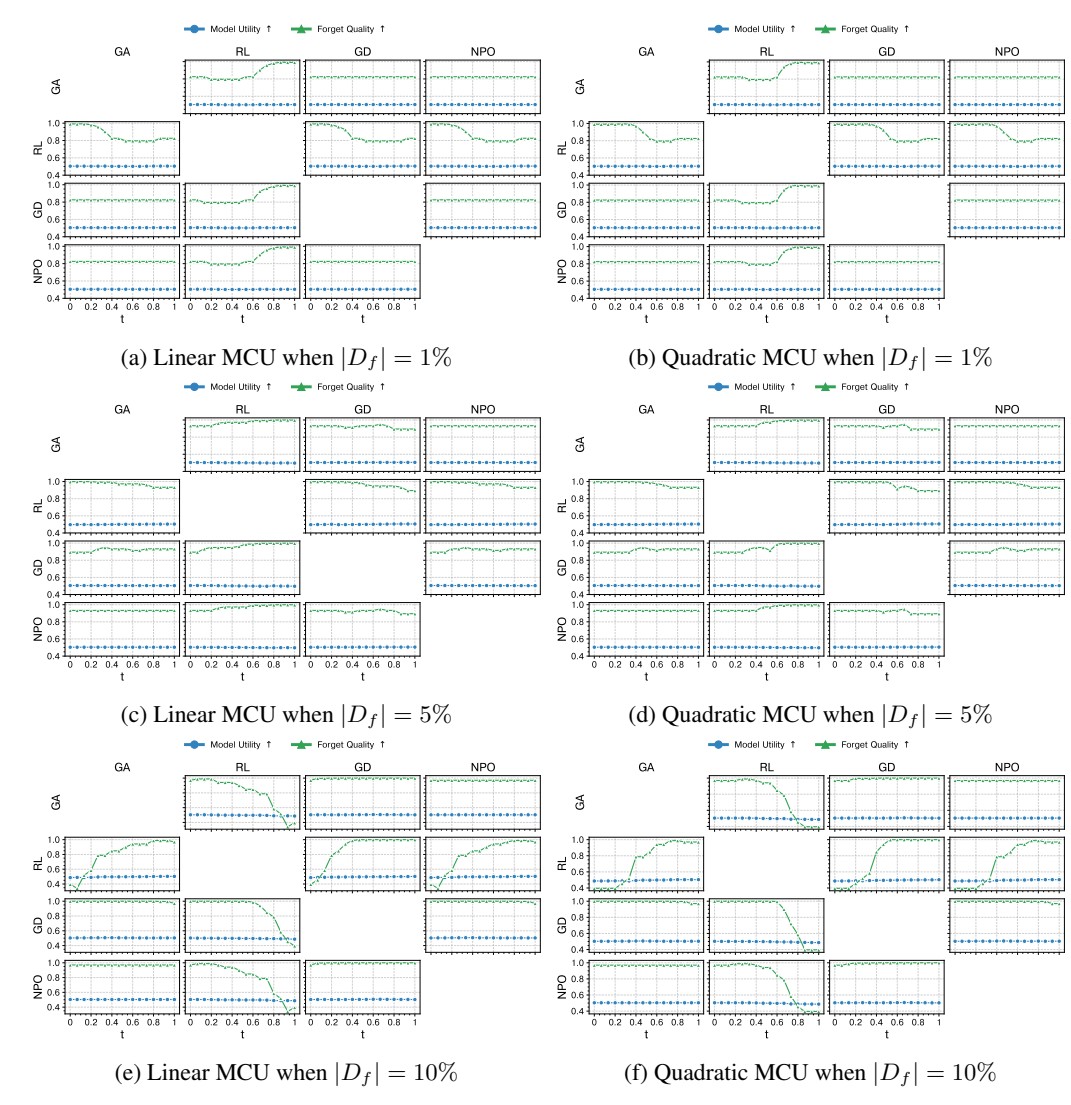

(a) Linear MCU when $|D_f| = 1\%$       (b) Quadratic MCU when $|D_f| = 1\%$

(c) Linear MCU when $|D_f| = 5\%$       (d) Quadratic MCU when $|D_f| = 5\%$

(e) Linear MCU when $|D_f| = 10\%$       (f) Quadratic MCU when $|D_f| = 10\%$

Figure 12: MCU under **Met** setting on **TOFU dataset**.

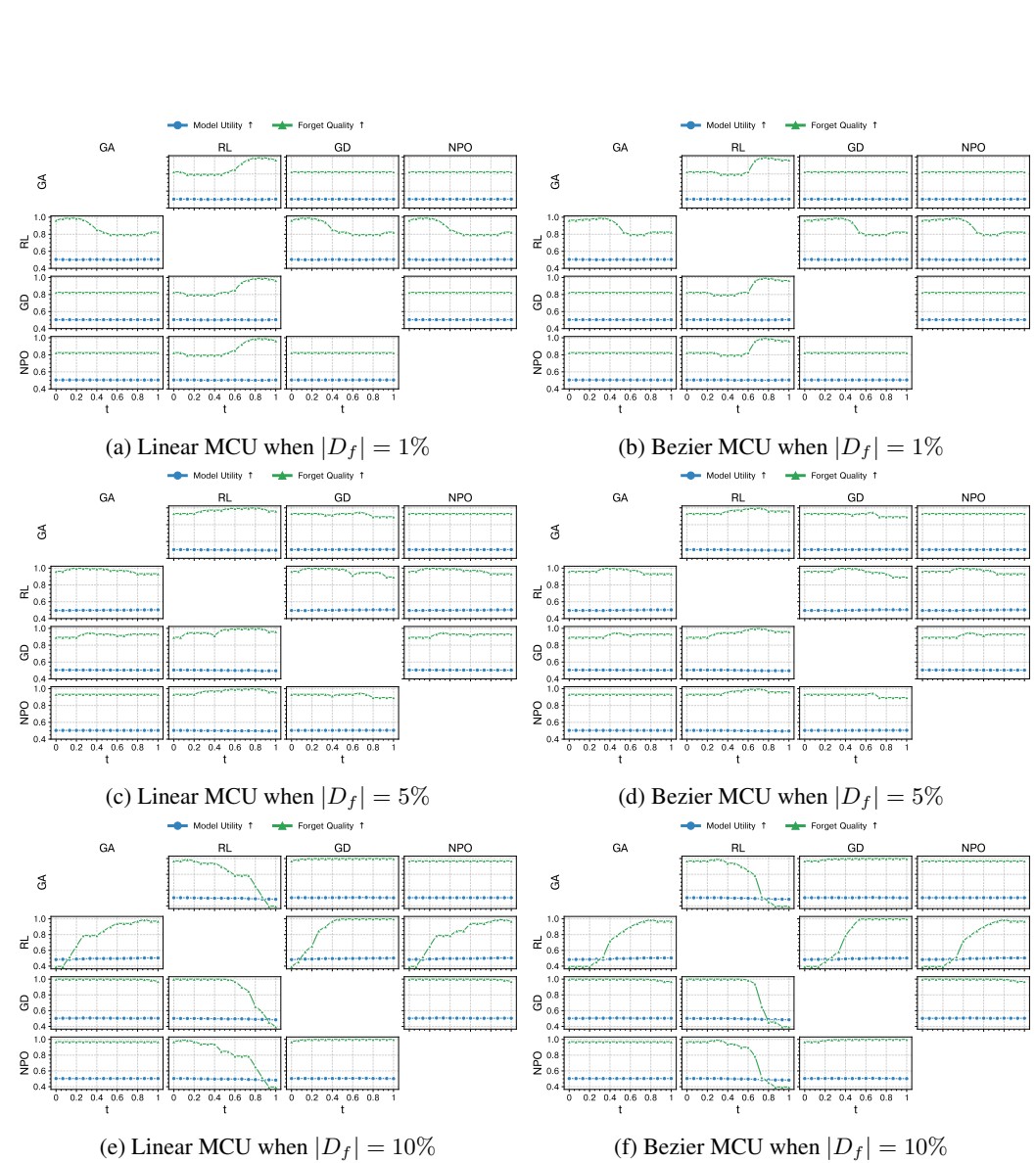

(a) Linear MCU when $|D_f| = 1\%$

(b) Bezier MCU when $|D_f| = 1\%$

(c) Linear MCU when $|D_f| = 5\%$

(d) Bezier MCU when $|D_f| = 5\%$

(e) Linear MCU when $|D_f| = 10\%$

(f) Bezier MCU when $|D_f| = 10\%$

Figure 13: MCU under **Met-CL** setting on **TOFU dataset**.

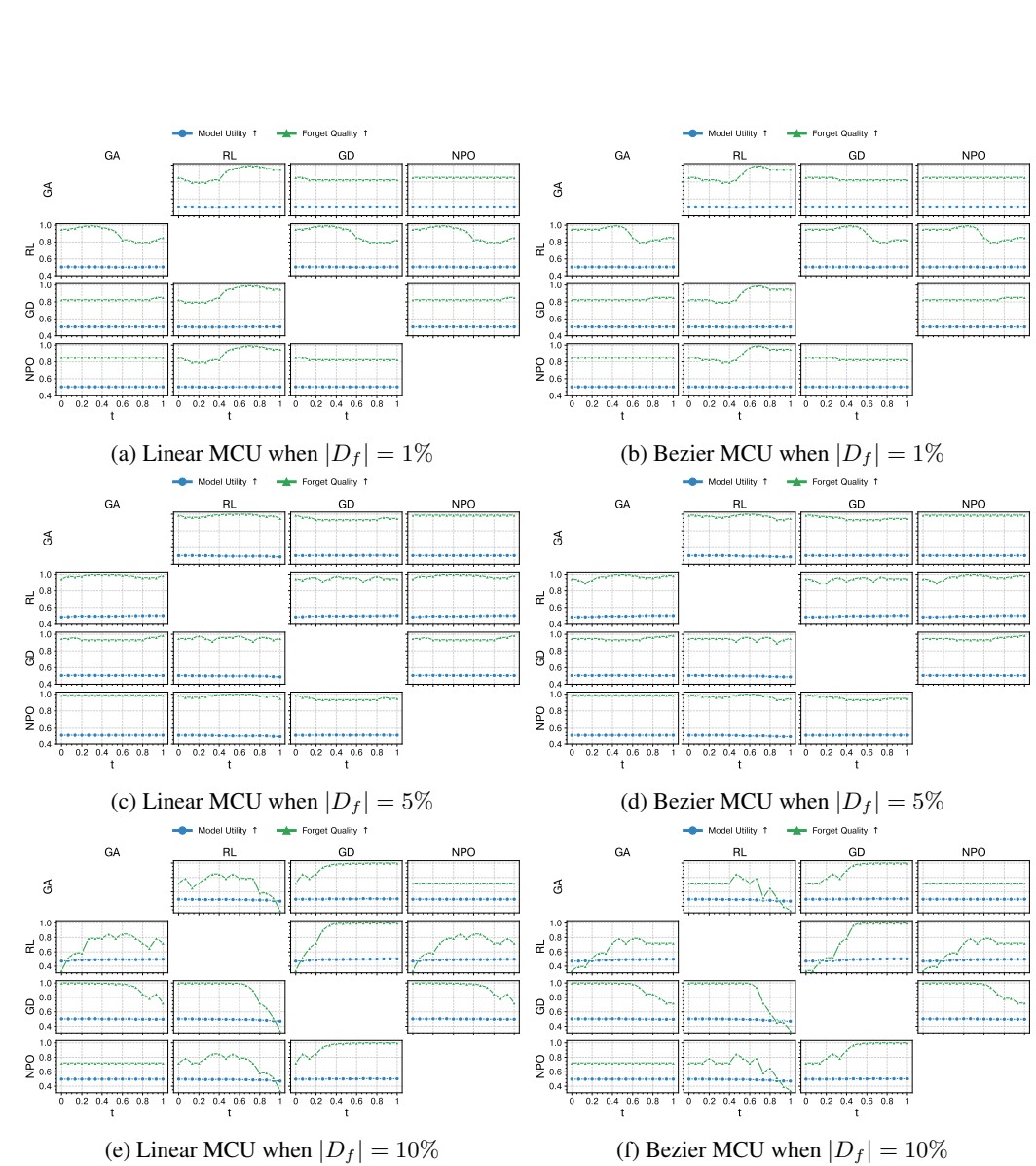

(a) Linear MCU when $|D_f| = 1\%$

(b) Bezier MCU when $|D_f| = 1\%$

(c) Linear MCU when $|D_f| = 5\%$

(d) Bezier MCU when $|D_f| = 5\%$

(e) Linear MCU when $|D_f| = 10\%$

(f) Bezier MCU when $|D_f| = 10\%$

Figure 14: MCU under **Met-SO** setting on **TOFU dataset**.

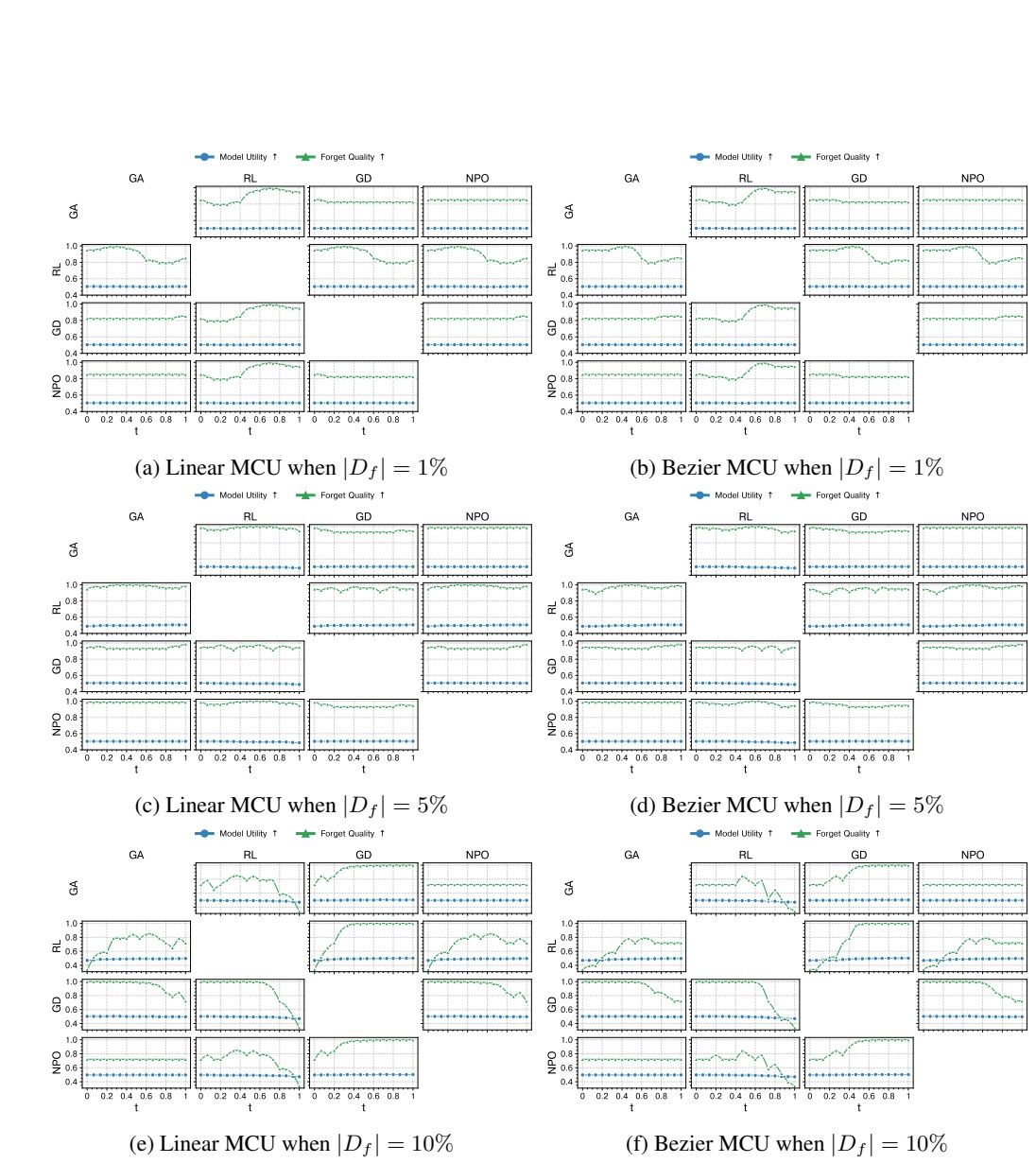

(a) Linear MCU when $|D_f| = 1\%$      (b) Bezier MCU when $|D_f| = 1\%$

(c) Linear MCU when $|D_f| = 5\%$      (d) Bezier MCU when $|D_f| = 5\%$

(e) Linear MCU when $|D_f| = 10\%$      (f) Bezier MCU when $|D_f| = 10\%$

Figure 15: MCU under **Met-CL-Non-CL** setting on **TOFU dataset**.

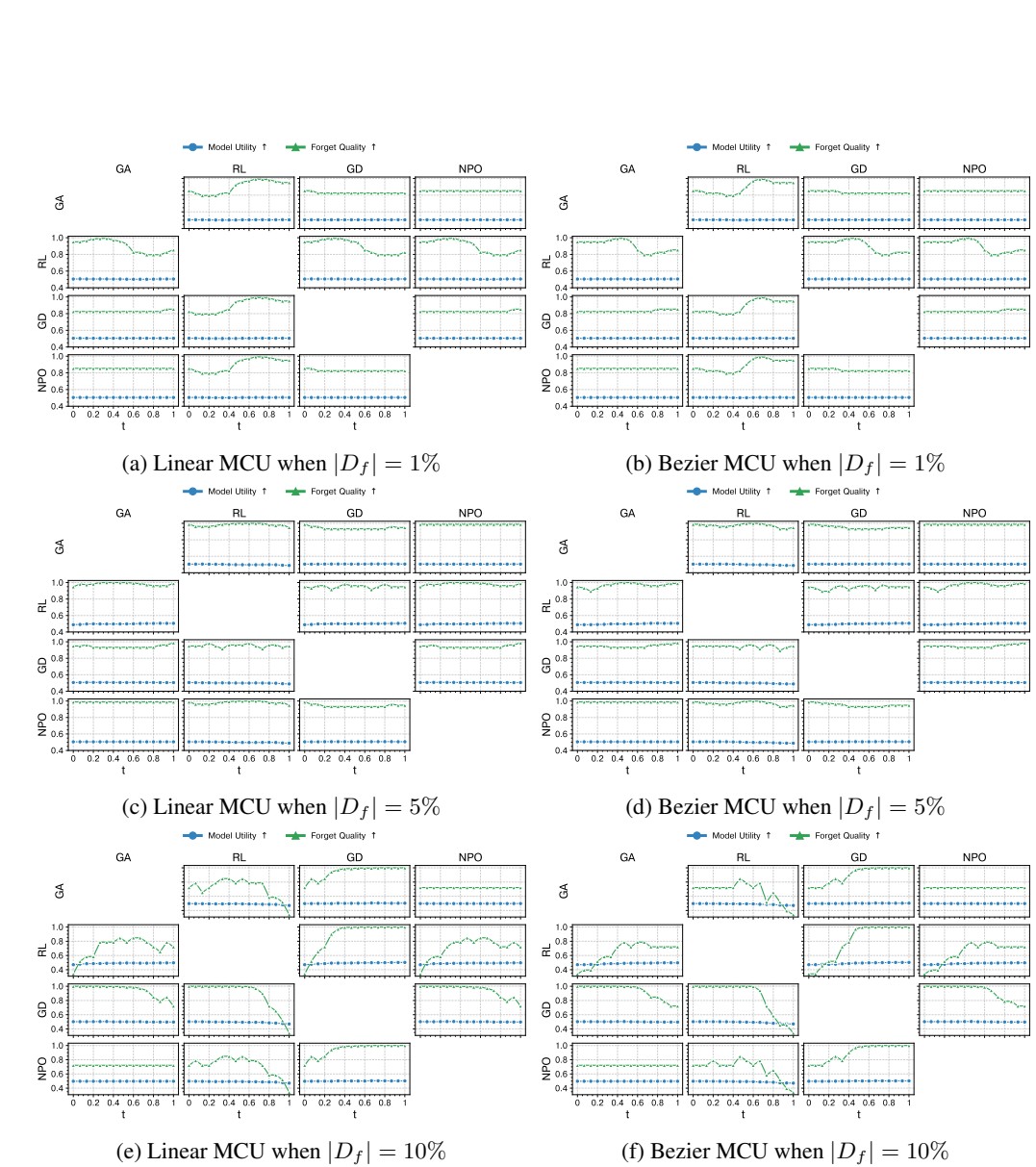

(a) Linear MCU when $|D_f| = 1\%$          (b) Bezier MCU when $|D_f| = 1\%$

(c) Linear MCU when $|D_f| = 5\%$          (d) Bezier MCU when $|D_f| = 5\%$

(e) Linear MCU when $|D_f| = 10\%$          (f) Bezier MCU when $|D_f| = 10\%$

Figure 16: MCU under **Met-FO-SO** setting on **TOFU dataset**.

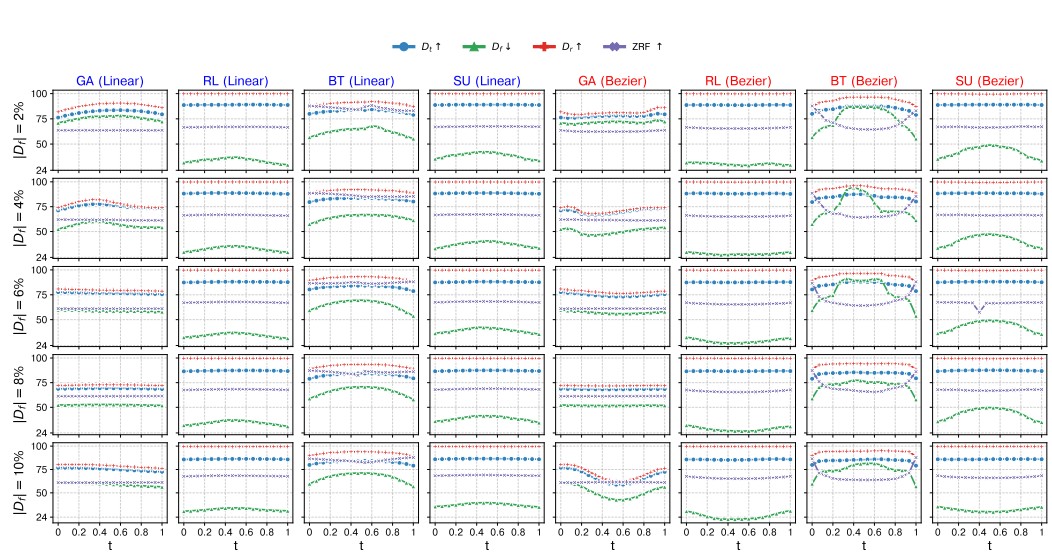

Figure 17: MCU under **Rand** setting on **classification dataset**.

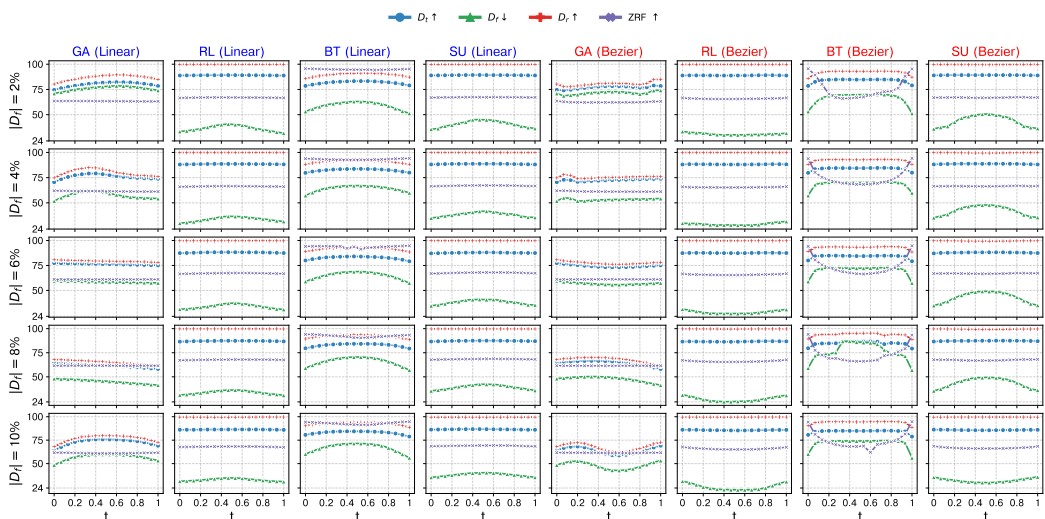

Figure 18: MCU under **Rand-CL** setting on **classification dataset**.

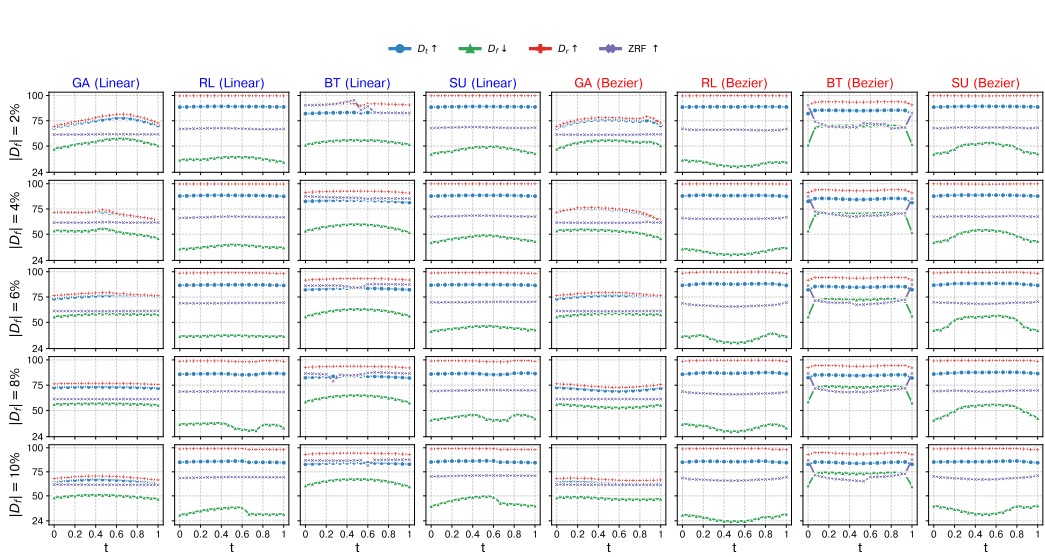

Figure 19: MCU under **Rand-SO** setting on **classification dataset**.

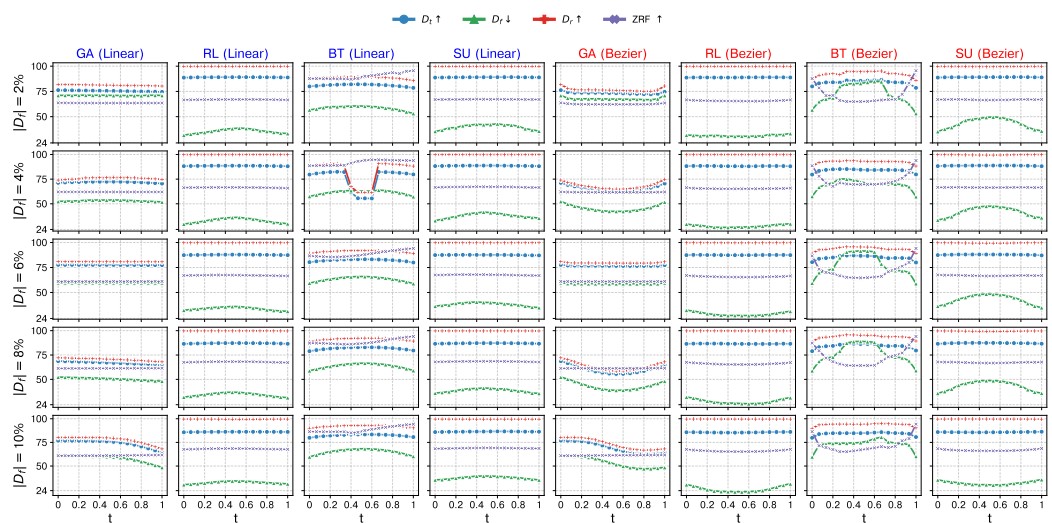

Figure 20: MCU under **CL-Non-CL** setting on **classification dataset**.

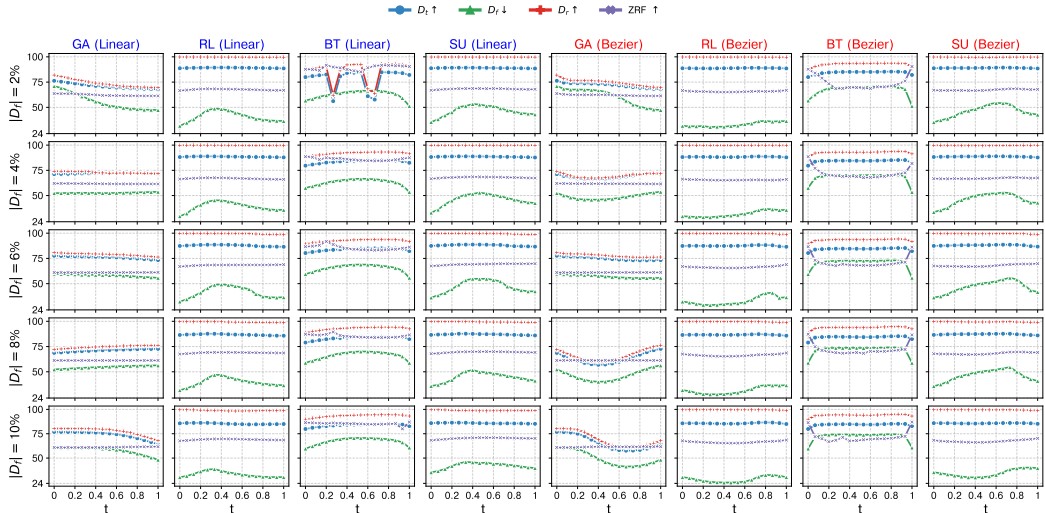

Figure 21: MCU under **FO-SO** setting on **classification dataset**.

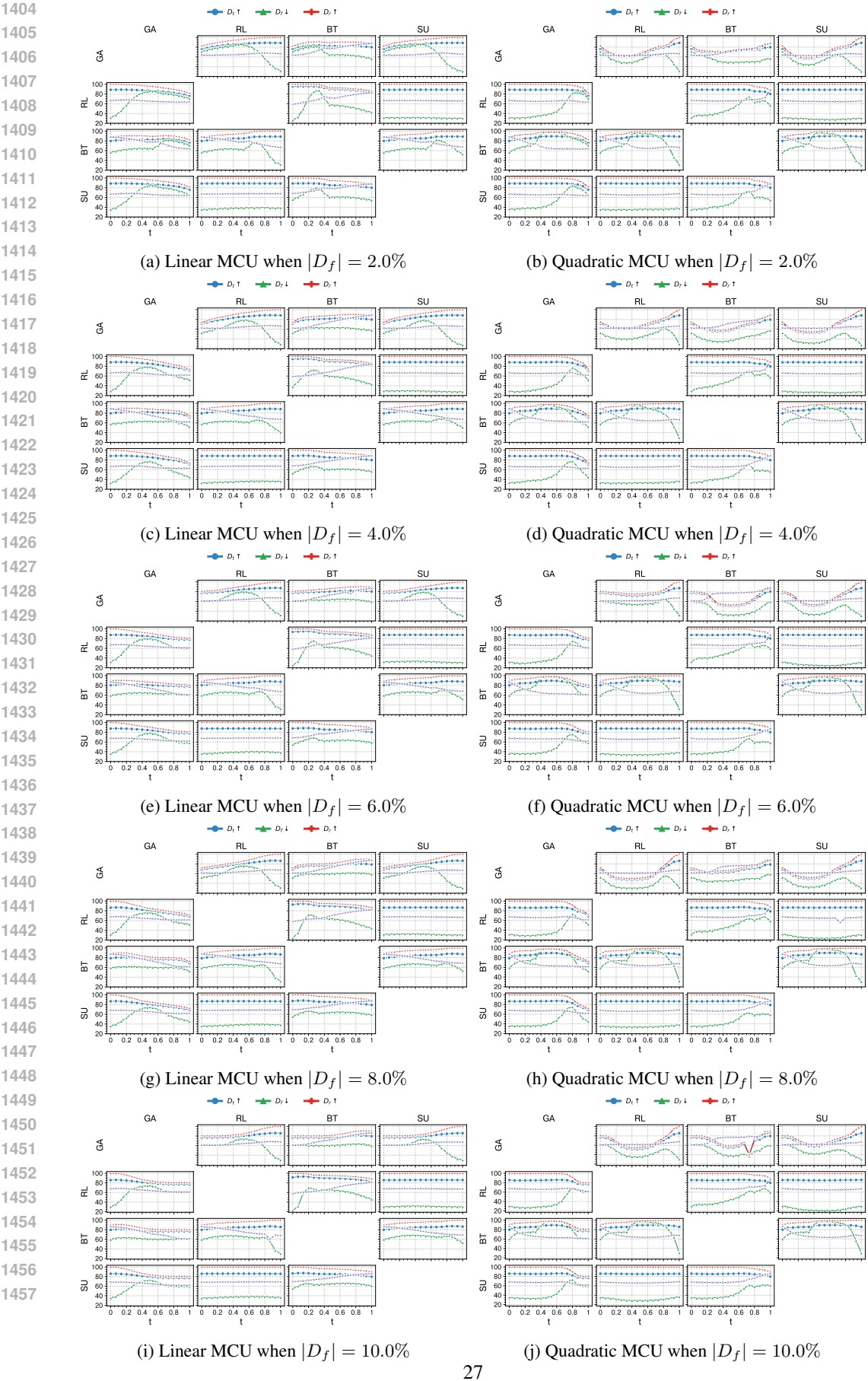

(a) Linear MCU when $|D_f| = 2.0\%$

(b) Quadratic MCU when $|D_f| = 2.0\%$

(c) Linear MCU when $|D_f| = 4.0\%$

(d) Quadratic MCU when $|D_f| = 4.0\%$

(e) Linear MCU when $|D_f| = 6.0\%$

(f) Quadratic MCU when $|D_f| = 6.0\%$

(g) Linear MCU when $|D_f| = 8.0\%$

(h) Quadratic MCU when $|D_f| = 8.0\%$

(i) Linear MCU when $|D_f| = 10.0\%$

(j) Quadratic MCU when $|D_f| = 10.0\%$

27

Figure 22: MCU under **Met** setting on **classification datasets**.

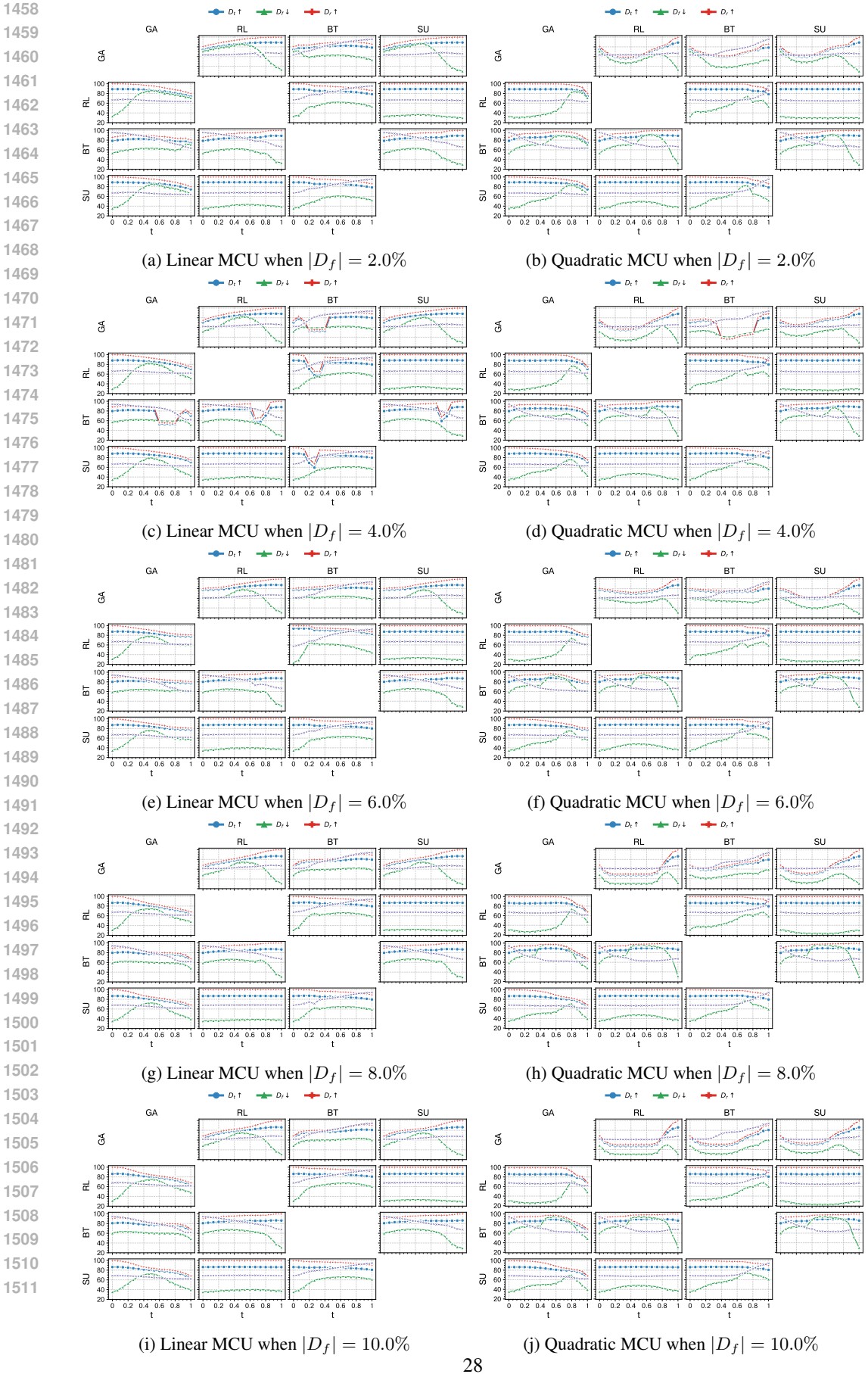

(a) Linear MCU when $|D_f| = 2.0\%$

(b) Quadratic MCU when $|D_f| = 2.0\%$

(c) Linear MCU when $|D_f| = 4.0\%$

(d) Quadratic MCU when $|D_f| = 4.0\%$

(e) Linear MCU when $|D_f| = 6.0\%$

(f) Quadratic MCU when $|D_f| = 6.0\%$

(g) Linear MCU when $|D_f| = 8.0\%$

(h) Quadratic MCU when $|D_f| = 8.0\%$

(i) Linear MCU when $|D_f| = 10.0\%$

(j) Quadratic MCU when $|D_f| = 10.0\%$

Figure 23: MCU under **Met-CL** setting on **classification datasets**.

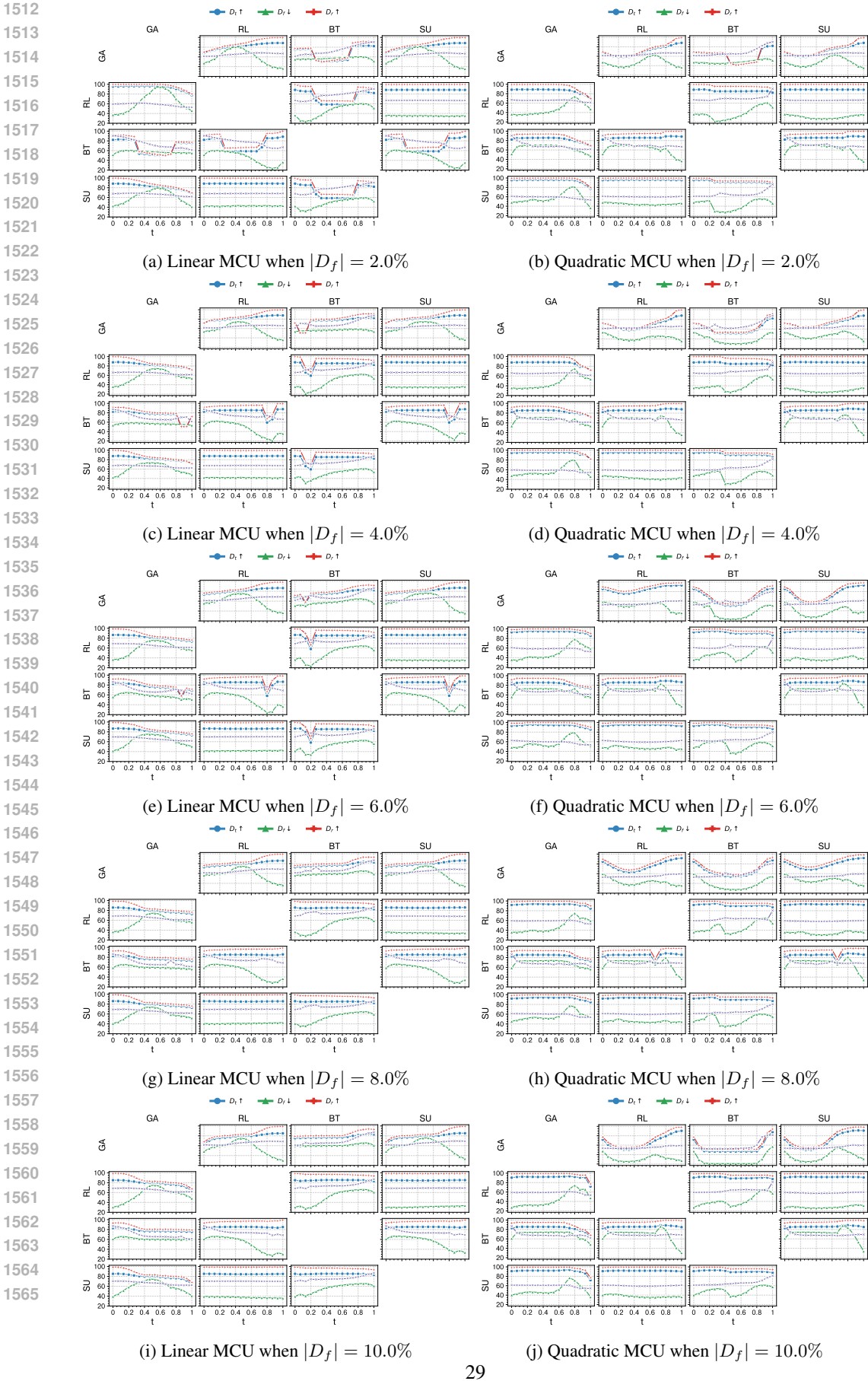

(a) Linear MCU when $|D_f| = 2.0\%$

(b) Quadratic MCU when $|D_f| = 2.0\%$

(c) Linear MCU when $|D_f| = 4.0\%$

(d) Quadratic MCU when $|D_f| = 4.0\%$

(e) Linear MCU when $|D_f| = 6.0\%$

(f) Quadratic MCU when $|D_f| = 6.0\%$

(g) Linear MCU when $|D_f| = 8.0\%$

(h) Quadratic MCU when $|D_f| = 8.0\%$

(i) Linear MCU when $|D_f| = 10.0\%$

(j) Quadratic MCU when $|D_f| = 10.0\%$

29

Figure 24: MCU under **Met-SO** setting on **classification datasets**.

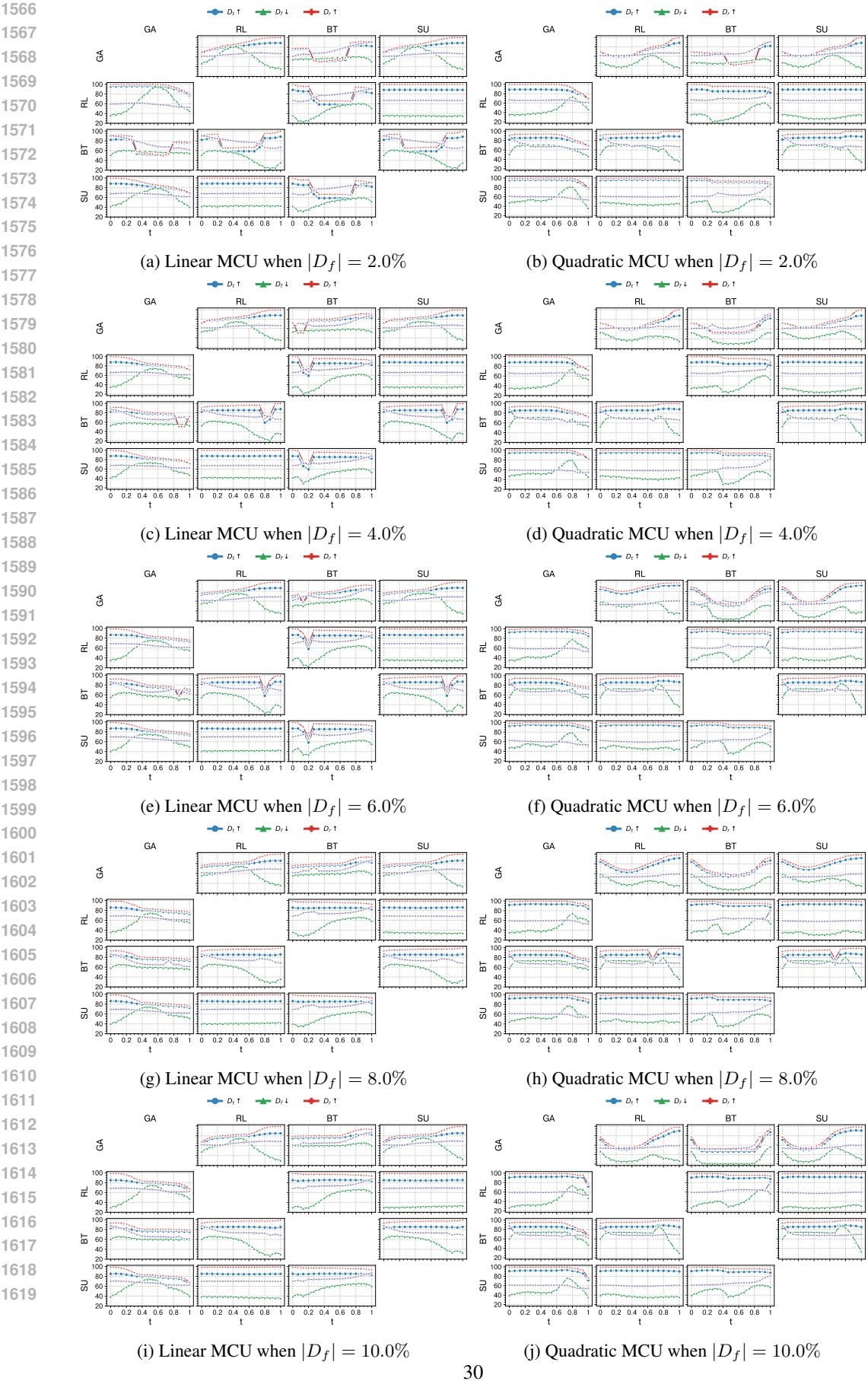

(a) Linear MCU when $|D_f| = 2.0\%$

(b) Quadratic MCU when $|D_f| = 2.0\%$

(c) Linear MCU when $|D_f| = 4.0\%$

(d) Quadratic MCU when $|D_f| = 4.0\%$

(e) Linear MCU when $|D_f| = 6.0\%$

(f) Quadratic MCU when $|D_f| = 6.0\%$

(g) Linear MCU when $|D_f| = 8.0\%$

(h) Quadratic MCU when $|D_f| = 8.0\%$

(i) Linear MCU when $|D_f| = 10.0\%$

(j) Quadratic MCU when $|D_f| = 10.0\%$

30

Figure 25: MCU under **Met-CL-Non-CL** setting on **classification datasets**.

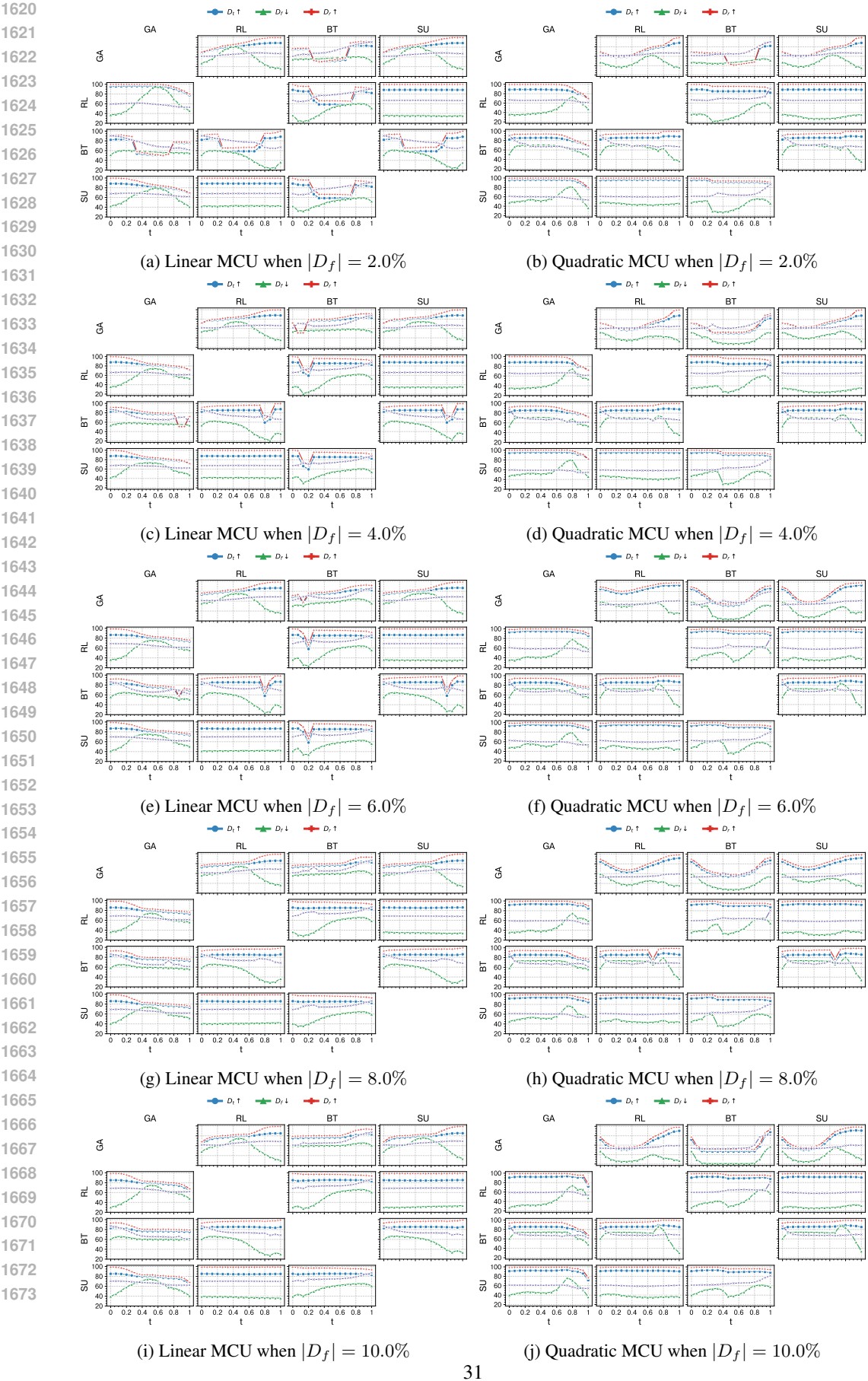

(a) Linear MCU when $|D_f| = 2.0\%$

(b) Quadratic MCU when $|D_f| = 2.0\%$

(c) Linear MCU when $|D_f| = 4.0\%$

(d) Quadratic MCU when $|D_f| = 4.0\%$

(e) Linear MCU when $|D_f| = 6.0\%$

(f) Quadratic MCU when $|D_f| = 6.0\%$

(g) Linear MCU when $|D_f| = 8.0\%$

(h) Quadratic MCU when $|D_f| = 8.0\%$

(i) Linear MCU when $|D_f| = 10.0\%$

(j) Quadratic MCU when $|D_f| = 10.0\%$

31

Figure 26: MCU under **Met-FO-SO** setting on **classification datasets**.