# OpenReview forum: "Mode Connectivity in Unlearning: A Loss Landscape Analysis of Machine Unlearning"
_ICLR.cc/2026/Conference — ICLR 2026 Conference Withdrawn Submission_

### Official Review · Reviewer_oRJb · 2025-10-21

**Soundness:** 2
**Presentation:** 2
**Contribution:** 2
**Rating:** 4
**Confidence:** 1

**Summary:**

The paper formalizes mode connectivity in unlearning. It contrasts MCU with standard mode connectivity, since MCU jointly constrains two loss landscapes and derives both endpoints from the same pretrained model. Experiments across datasets, unlearning methods, and training paradigms show MCU often emerges, with prevalence influenced by method choise, taks complexty, forget-set size, curriculum learning, and second-order optijmization.

**Strengths:**

- Clear MCU definition with retain/forget inequalities and curve parametrization.
- Systematic study of training dynamics, extending from mode-connectivity evaluations.
- MCU-guided ensembling yields smoother landscapes and improved robustness to relearning.

**Weaknesses:**

- Many findings are illustrated in the appendix but not aggregated into compact quantitative summaries in the main text.
- Quadratic MCU often fails to optimize for certain methods, like BT, with limited diagnosis.

**Questions:**

- How sensitive are MCU-ensemble gains to the number/placement of sampled points along the path and to the averaging scheme?

---

### Official Review · Reviewer_ksW1 · 2025-10-28

**Soundness:** 3
**Presentation:** 3
**Contribution:** 2
**Rating:** 4
**Confidence:** 2

**Summary:**

This paper studies mode connectivity and its implications on Unlearning. The paper starts by extending the definition of mode connectivity for unlearning. The paper analyzes the loss landscape of machine unlearning and properties of different machine unlearning methods based on mode connectivity, such as generalization, robustness to attacks. The paper also analyzes similarity between unlearning methods and the unlearning difficulty of the task itself.

**Strengths:**

The paper extends mode connectivity under the lens of machine unlearning. The paper provides a novel extension to the mode connectivity literature extending its basic definition to the setting of unlearning. The paper also monitors additional parameters, which affect the training dynamics, studies around these parameters were absent in previous works.

This work continues to provide a substantial list of experimental results that study a lot of important questions in the field of unlearning. The results provide empirical evidence for the robustness against attacks of different solutions found by unlearning algorithms. Interestingly the paper also provides a criterium through mode connectivity for unlearning of the unlearning difficulty.

**Weaknesses:**

The papers weaknesses for me lie in the extension of the notion of mode connectivity from simple minimization (model training) to machine unlearning.

The maximization of the loss on the forget set is an arbitrary solution that has been applied to emulate a model that has never seen the samples in the forget set. Therefore while for the minimization problem the condition for what proper mode connectivity is, is clear in the setting of unlearning it is not. In fact there is a recent surge of works that argue against loss maximization on the forget set.

This also brings me to my next point how does the MCU manifold guarantee generalization as stated in 6.2? In the simpler MC manifold the arguement was simpler and does not extend here directly due to the additional maximization condition on the error. For example consider removing a forget set that is representative of the true distribution (even though we don't know a priori obviously) your imposed condition would in fact guarantee poor generalization under some forget sets.

**Questions:**

My questions revolve around the weaknesses.

1. Would it not make more sense to study the connectivity between the modes of unlearned methods and an oracle model, that has never seen the forget? Would it make sense to quantify the surplus of the loss of the path $\phi(t)$ in equation (2) that is needed to connect the two modes?

2. Why should we do maximization on the forget, as I said before this choice seems arbitrary to me. Would it make sense to instead measure KL divergence [1] between the model and a proxy of the unlearned model, or KL between the forget set and a validation set?

[1] Attribute-to-Delete: Machine Unlearning via Datamodel Matching Kristian Georgiev, Roy Rinberg, Sung Min Park, Shivam Garg, Andrew Ilyas, Aleksander Madry, Seth Neel

---

### Official Review · Reviewer_DXr2 · 2025-10-30

**Soundness:** 2
**Presentation:** 1
**Contribution:** 2
**Rating:** 2
**Confidence:** 4

**Summary:**

The authors examine the mode connectivity of various machine unlearning methods. That is, are there simple, low-loss paths between different models that have undergone some machine unlearning procedure? They look for patterns in the mode connectivity behavior of different unlearning methods (e.g. do the loss landscapes of second order unlearning methods behave similarly)

**Strengths:**

The idea that smoother loss landscapes could indicate better unlearning is intuitive and worth exploring. I appreciate that the authors tried *a lot* of unlearning methods

The idea of ensembling along a mode connectivity path and understanding how the spikiness of the loss landscape relates to the difficulty of the unlearning task both seem like promising directions.

**Weaknesses:**

- The takeaway from this paper seems to be "this is really messy and it's hard to say what's going on." To be fair, it may be worth knowing that it's messy, but this paper itself is *also* quite messy so it's hard to know what to think.
- Given the messiness of the results, I don't know that I trust the generalized claims about e.g. second-order methods. It seems like sometimes you can find a nice mode connectivity path but, if you can't, it could be the method, it could be the dataset, it could be the way you search for the path, it could be the random seed, it could be a bug...
- In general, I tried to judge the paper on its merits and not get hung up on this, but there are *a lot* of spelling mistakes and grammatical issues throughout. Please proofread! Or have an LLM do it!

Suggestions:
- Figure 1 seems to imply that the loss landscape is quadratic, which is not what I think you're trying to say. May be better without the "landscape" in the figure?

**Questions:**

I understand that the figures (e.g. figure 5) are supposed to be showing the mode connectivity on some trajectory between models, but...I don't think I actually understand what is happening here. Can you explain this figure?

---

### Official Review · Reviewer_LD6j · 2025-11-03

**Soundness:** 3
**Presentation:** 2
**Contribution:** 2
**Rating:** 4
**Confidence:** 4

**Summary:**

This paper studies mode connectivity (the phenomenon where different parameters can in some cases define a continuous path of well-performing models) in the context of machine unlearning. The paper first alters the definition of mode connectivity, from only requiring a path of parameters that achieve low loss to requiring a parameters that achieve low loss on the retain set, and high loss on the target/forget set. With this definition in hand, the paper investigates mode connectivity between different unlearning algorithms on a variety of datasets, and reports its findings.

**Strengths:**

- The application of mode connectivity analysis to unlearning is novel to my knowledge
- The authors explain the details of their procedure very well - as a reader it's very clear which experiments are being done and why
- The experiments are thorough, and completed on a variety of unlearning datasets which makes the results more generalizable

**Weaknesses:**

The takeaways/high-level message from the paper are not entirely clear, for example:

- In RQ1, the answer to most of the subquestions seems to be "it depends on the dataset and the exact setup," which is fine in some cases, but the paper should try to form an explanation of why the results differ across different setups/what about the setup makes conclusions differ
- The paper claims (in RQ2) that MCU is a useful tool for studying unlearning, but most of the results in the corresponding section (S6) are reproductions of known results/phenomena in unlearning, and not discoveries of new behavior - more importantly, it's unclear why MCU is a natural/right way to discover this behavior.
- Section 6 in general looks like the beginning of an interesting investigation but the results are not fleshed out enough to stand on their own, in my opinion.

**Questions:**

Setup:
- When computing a quadratic path for MCU, is the loss on the forget set taken into account too, or just the retain set/test set?
- Could this choice have any bearing on the result?
- Are there other mode connectivity algorithms, and do you expect the results to vary much as you change the type of mode connectivity?

Experiments:
- Did you try this analysis for "ground-truth" unlearning (i.e., just training on the retain set and not the forget set)?
- Is there mode connectivity between this and the other methods studied?
- Does mode connectivity to "ground-truth unlearning" indicate something about how robust/reliable a method is?

---

### Note · Authors · 2025-12-30

I have read and agree with the venue's withdrawal policy on behalf of myself and my co-authors.